# The PMIP4 contribution to CMIP6 - Part 3: the Last Millennium, Scientific Objective and Experimental Design for the PMIP4 *past1000* simulations

5 Johann H. Jungclaus[1], Edouard Bard[2], Mélanie Baroni[2], Pascale Braconnot[3], Jian Cao[4], Louise P. Chini[5], Tania Egorova[6,7], Michael Evans[8], J. Fidel González-Rouco[9], Hugues Goosse[10], George C. Hurtt[5], Fortunat Joos[11], Jed O. Kaplan[12], Myriam Khodri[13], Kees Klein Goldewijk[14,15], Natalie Krivova[16], Allegra N. LeGrande[17], Stephan J. Lorenz[1], Jürg Luterbacher[18,19], Wenmin Man[20], Amanda C. Maycock[21], Malte Meinshausen[22,23], Anders Moberg[24], Raimund Muscheler[25], Christoph Nehrbass-Ahles[11], Bette I. Otto-Bliesner[26], Steven J. Phipps[27], Julia Pongratz[1], Eugene Rozanov[6,7], Gavin A. Schmidt[17], Hauke Schmidt[1], Werner Schmutz[6], Andrew Schurer[28], Alexander I. Shapiro[16], Michael Sigl[29,30], Jason E. Smerdon[31], Sami K. Solanki[16], Claudia Timmreck[1], Matthew Toohey[32], Ilya G. Usoskin[33], Sebastian Wagner[34], Chi-Ju Wu[16], Kok Leng Yeo[16], Davide Zanchettin[35], Qiong Zhang[24], and Eduardo Zorita[34]

[1]Max Planck Institut für Meteorologie, Hamburg, Germany
[2]CEREGE, Aix-Marseille University, CNRS, IRD, College de France, Technopole de l'Arbois, 13545 Aix-en-Provence, France
[3]Laboratoire des Sciences du Climat et de l'Environnement, LSCE/ IPSL, CEA –CNRS-UVSQ, Université Paris-Saclay, F-91191 Gif-sur-Yvette, France
[4]Earth System Modeling Center, Nanjing University of Information Science and Technology, Nanjing 210044, China
[5]Department of Geographical Sciences, University of Maryland, College Park, MD 20742
[6]Physikalisch-Meteorologisches Observatorium Davos and World Radiation Center (PMOD/WRC), Davos, Switzerland.
[7]Institute for Atmospheric and Climate Science, ETH Zurich, Switzerland
[8]Dept of Geology and Earth System Science Interdisciplinary Center, University of Maryland, College Park, MD 20742 USA.
[9]Dept. of Astrophysics and Atmospheric Sciences. IGEO (UCM-CSIC). Universidad Complutense de Madrid, 28040 Madrid, Spain.
[10]ELI/TECLIM, Université Catholique de Louvain, Belgium
[11]Climate and Environmental Physics, Physics Institute and Oeschger Centre for Climate Change Research, University of Bern, Bern, Switzerland
[12]Institute of Earth Surface Dynamics, University of Lausanne, Switzerland
[13]Laboratoire d'Océanographie et du Climate, Sorbonne Universités, UPMC Université Paris 06, IPSL, UMR CNRS/IRD/MNHN, F-75005 Paris, France
[14]Copernicus Institute of Sustainable Development, Utrecht University, Utrecht, The Netherlands
[15]PBL Netherlands Environmental Assessment Agency, The Hague/Bilthoven, The Netherlands
[16]Max-Planck-Institut für Sonnensystemforschung, Göttingen, Germany
[17]NASA Goddard Institute for Space Studies, 2880 Broadway, New York, USA
[18]Department of Geography, Climatology, Climate Dynamics and Climate Change, Justus Liebig University Giessen, Germany
[19]Centre for International Development and Environmental Research, Justus Liebig University Giessen, Germany
[20]LASG Institute of Atmospheric Physics, Chinese Academy of Sciences, Beijing, China
[21]School of Earth and Environment, University of Leeds, Leeds, UK
[22]Australian-German Climate & Energy College, the University of Melbourne, Australia
[23]Potsdam Institute for Climate Impact Research, Potsdam, Germany
[24]Department of Physical Geography and Bolin Centre for Climate Research, Stockholm University, Sweden
[25]Department of Geology, Lund University, Lund, Sweden
[26]National Center for Atmospheric Research, Boulder, Colorado 80305, USA.
[27]Institute for Marine and Antarctic Studies, University of Tasmania, Hobart, Tasmania, Australia
[28]GeoSciences, University of Edinburgh, Edinburgh, UK
[29]Laboratory of Environmental Chemistry, Paul Scherrer Institute, 5232 Villigen, Switzerland
[30]Oeschger Centre for Climate Change Research, University of Bern, 3012 Bern, Switzerland
[31]Lamont-Doherty Earth Observatory of Columbia University, Palisades, NY, USA
[32]GEOMAR Helmholtz Centre for Ocean Research Kiel, Germany
[33]Space Climate Research Group and Sodankylä Geophysical Observatory, University of Oulu, Finland
[34]Institute for Coastal Research, Helmholtz-Zentrum Geesthacht, Geesthacht, Germany
[35]Department of Environmental Sciences, Informatics and Statistics, University of Venice, Mestre, Italy

*Correspondence to:* Johann Jungclaus (johann.jungclaus@mpimet.mpg.de)

**Abstract.** The pre-industrial millennium is among the periods selected by the Paleoclimate Model Intercomparison Project (PMIP) for experiments contributing to the sixth phase of the Coupled Model Intercomparison Project (CMIP6) and the fourth phase of PMIP (PMIP4). The *past1000* transient simulations serve to investigate the response to (mainly) natural forcing under background conditions not too different from today, and to discriminate between forced and internally generated variability on interannual to centennial time scales. This manuscript describes the motivation and the experimental set-ups for the PMIP4-CMIP6 *past1000* simulations, and discusses the forcing agents: orbital, solar, volcanic, land-use/land-cover changes, and variations in greenhouse gas concentrations. The *past1000* simulations covering the pre-industrial millennium from 850 Common Era (CE) to 1849 CE have to be complemented by *historical* simulations (1850 to 2014 CE) following the CMIP6 protocol. The external forcings for the *past1000* experiments have been adapted to provide a seamless transition across these time periods. Protocols for the *past1000* simulations have been divided into three tiers. A default forcing data set has been defined for the "tier-1" (the CMIP6 *past1000*) experiment. However, the PMIP community has maintained the flexibility to conduct coordinated sensitivity experiments to explore uncertainty in forcing reconstructions as well as parameter uncertainty in dedicated "tier-2" simulations. Additional experiments ("tier-3") are defined to foster collaborative model experiments focusing on the early instrumental period and to extend the temporal range and the scope of the simulations. This manuscript outlines current and future research foci and common analyses for collaborative work between the PMIP and the observational communities (reconstructions, instrumental data).

Keywords: Climate and Earth system modelling, CMIP6, PMIP, last millennium, natural forcing

## 1 Introduction

Based on a vast collection of proxy and observational data sets, the Common Era (CE; approximately the last 2000 years) is the best-documented interval of decadal- to centennial-scale climate change in Earth's history (PAGES2K Consortium, 2013, 2014; Masson-Delmotte et al., 2013). Climate variations during this period have left their traces on human history, such as the documented impacts of the Medieval Climate Anomaly (MCA) and the Little Ice Age (LIA) (e.g., Pfister and Brázdil, 2006; Büntgen et al., 2016; Xoplaki et al., 2016; Camenisch et al., 2016). Nevertheless, there is still debate regarding the relative contribution of internal variability and external forcing factors to natural fluctuations in the Earth's climate system and how they compare to the present anthropogenic global warming (Masson-Delmotte et al., 2013). This is particularly acute for regional and sub-continental scales, where spatially heterogeneous variability modes potentially impact the climate signal (e.g., PAGES2k-PMIP3 Group, 2015; Luterbacher et al., 2016; Gagen et al., 2016). Simulations covering the recent past can thus provide context for the evolution of the modern climate system and for the expected changes during the coming decades and centuries. Furthermore, they can help to identify plausible mechanisms underlying palaeoclimatic observations and reconstructions.  Here, we describe and discuss the forcing boundary conditions and experimental protocol for the *past1000* simulations covering the pre-industrial millennium (850 to 1849 CE) as part of the fourth phase of the Paleoclimate Model Intercomparison Project (PMIP4, Kageyama et al., 2016) and the sixth phase of the Coupled Model Intercomparison Project (CMIP6, Eyring et al., 2016). We emphasize, that the *past1000* simulations must be complemented by *historical* simulations for 1850 to 2014 CE following the CMIP6 protocol and applying the CMIP6 external forcing for the industrial period (Eyring et al., 2016 and references therein).

Simulations of the CE have applied models of varying complexity. Crowley (2000) and Hegerl et al. (2006) used Energy Balance Models to study the surface temperature response to changes in external forcing, particularly solar, volcanic and greenhouse gas concentrations (GHG). Earth System Models (ESM) of Intermediate Complexity (e.g., Goosse et al., 2005) have been used to perform long integrations or multiple (ensemble) simulations requiring relatively small amounts of computer resources. Finally, coupled Atmosphere Ocean General Circulation Models (AOGCM) and comprehensive ESMs have enabled the community to gain further insights into internally generated and externally-forced variability, investigating climate dynamics, modes of variability (e.g., González-Rouco et al., 2003, 2006; Raible et al., 2014; Ortega et al., 2015; Zanchettin et al., 2015; Landrum et al., 2013) and regional processes in greater detail (Goosse et al., 2006, 2012; PAGES2k-PMIP3 Group, 2015; Coats et al., 2015; Luterbacher et al., 2016). They have also allowed individual groups to study specific components of the climate system, such as the carbon cycle (Jungclaus et al., 2010; Lehner et al., 2015; Chikamoto et al., 2016), or aerosols and short-lived gases (e.g., Stoffel et al., 2015). Recent increases in computing power have made it feasible to carry out millennial-scale ensemble simulations with comprehensive ESMs (e.g., Jungclaus et al., 2010; Otto-Bliesner et al., 2016).  Ensemble approaches are extremely beneficial as a means of separating and quantifying simulated internal variability and the responses to changes in external forcing, under the assumption that the simulation variance within the ensemble is a reasonable estimate of the unforced variability of the actual climate system (e.g., Deser et al., 2012; Stevenson et al., 2016).

The *past1000* experiment was adopted as a standard experiment in the third phase of PMIP (PMIP3, Braconnot et al., 2012), which was partly embedded within the fifth phase of CMIP (CMIP5, Taylor et al., 2012). This was an important step as it encouraged modelling groups to use the same climate models for future scenarios and for palaeoclimate simulations, instead of stripped-down or low-resolution versions. Using the same state-of-the-art ESMs to simulate both past and future climates allows palaeoclimate data to be used to evaluate the same models that are, in turn, employed to generate future climate projections (Schmidt et al., 2014). The PMIP3 *past1000* experiments were based on a common protocol describing a variety

of suitable forcing boundary conditions (Schmidt et al., 2011; 2012). Moreover, a common structure of the CMIP5 output facilitated multi-model analyses, comparisons with reconstructions and connections to future projections (e.g., Bothe et al., 2013; Smerdon et al., 2015; PAGES2k-PMIP3 Group, 2015; Cook et al., 2015). Several studies have also addressed variations and responses of the carbon cycle (e.g., Brovkin et al., 2010; Lehner et al., 2015; Keller et al., 2015; Chikamoto et al., 2016). Last-millennium related contributions to several chapters of Assessment Report 5 of the Intergovernmental Panel on Climate Change (IPCC-AR5) (Masson-Delmotte et al., 2013; Flato et al., 2013; Bindoff et al., 2013) highlighted the value of the *past1000* multi-model ensemble.

Further progress is expected for CMIP6 and PMIP4. Models with higher spatial resolution will be available for long-term paleo simulations, which has the potential to improve the representation of mechanisms controlling regional variability and to alleviate biases in the mean state (e.g. Milinski et al., 2016). Newly added model components, for example interactive chemistry and aerosol microphysics, will allow for more explicit representation of forcing-related processes in some models (e.g., LeGrande et al., 2016), and, as we outline below, improvements in forcing reconstructions regarding their accuracy and complexity will potentially lead to improved quality in comparative model/data studies. In addition, more stringent protocols for experimental set-ups (e.g., an identical forcing data set for the "tier-1" experiment) and output data are implemented in the CMIP6 process, providing an improved for basis multi-model studies. The CMIP6 protocols also ensure a better interaction between related MIPs. For example, the PMIP4 *past1000* experiment is closely related to the more process-oriented suite of simulations in the Model Intercomparison Project on the climatic response to Volcanic forcing (VolMIP, Zanchettin et al., 2016).

The PMIP working group on the climate evolution over the last 2000 years (WG Past2K) is closely cooperating with the PAGES (Past Global Changes) 2k Network promoting regional reconstructions of climate variables and modes of variability. Collaborative work has focused on reconstruction-model intercomparison (e.g. Bothe et al., 2013; Moberg et al., 2015; PAGES2k-PMIP3 Group, 2015) and assessment of modes of variability (e.g. Raible et al., 2014). Integrated assessment of reconstructions and simulations has led to progress in model evaluation and process understanding (e.g. Lehner et al., 2013; Sicre et al., 2013; Jungclaus et al., 2014; Man et al., 2014; Man and Zhou, 2014). The increasing number of available simulations and reconstructions has also created a need for development of new statistical modelling approaches dedicated to model-data comparison analysis (e.g. Sundberg et al., 2012; Barboza et al., 2014; Tingley et al., 2015; Bothe et al., 2015). The combination of real-world proxies with simulated "pseudo" proxies has improved the interpretation of the reconstructions (e.g. Smerdon, 2012) and helped to provide information for the selection of proxy sites and numbers (Wang et al., 2015; Zanchettin et al., 2015; Smerdon et al., 2016; Hind et al., 2012; Lehner et al., 2012; Ortega et al., 2015). Despite significant advances in our ability to simulate reconstructed past changes, challenges still remain; for example, regarding hydroclimatic changes in the last millennium (Anchukaitis et al., 2010; Ljungqvist et al., 2016). Documenting progress and the status of achievements and challenges in the multi-model context is a major goal of PMIP as the community embarks on a new round of Model Intercomparison Projects.

This paper is part of a suite of five manuscripts documenting the PMIP contributions to CMIP6. Kageyama et al. (2016) provide an overview on the five selected time periods and the experiments. More specific information are given in the contributions for the mid-Holocene (*midHolocene*) and the previous interglacial (*lig127k*) by Otto-Bliesner et al. (2016), for the last glacial maximum (*lgm*) by Kageyama et al. (2017), and for the mid-Pliocene warm period (*midPliocene*) by Haywood et al. (2016), and the present manuscript on the last millennium (*past1000*). PMIP has adopted the CMIP6 categorization where the highest-priority experiments are classified as "tier-1", whereas additional sensitivity experiments or dedicated studies are "tier-2" or "tier-3". The standard experiments for the five periods are all ranked "tier-1". Modelling groups are not obliged to run all PMIP4-CMIP6 experiments. It is mandatory, however, for all participating groups to run at least one of the experiments that were run in previous phases of PMIP (i.e., *midHolocene* or *lgm*).

Our *past1000* manuscript is organized as follows. In section 2, we review the major forcing agents for climate evolution during the CE in the light of previous simulations of the past. Section 3 describes the experimental protocols for the tier-1 to tier-3 categorized experiments. Section 4 describes the derivations and the characteristics of the forcing boundary conditions. Section 5 discusses the relations between the PMIP experiments and the overarching research questions of CMIP6 and links to other MIPs. Section 6 provides a concluding discussion.

## 2 Drivers of climate variations during the CE

The major forcing agents during the pre-industrial millennium are changes in orbital parameters, solar irradiance, stratospheric aerosols of volcanic origin, and greenhouse gas (GHG) concentrations. Additional anthropogenic impacts arise from aerosol emissions and changes in land-surface properties as a result of land use (e.g. Pongratz et al., 2009; Kaplan et al., 2011). External drivers affect the climate system in several ways, ranging from millennial-scale trends, such as those induced by changing orbital parameters, to the response of relatively short-lived disturbances of the radiative balance, as in the case of volcanic activity. Additionally, feedbacks internal to the climate system may amplify, delay, or prolong the effect of forcing (e.g., Shindell et al., 2001; Swingedouw et al., 2011; Zanchettin et al., 2012). The PMIP4 experiments will revisit the questions regarding the relative role of external drivers using updated forcing datasets and a new generation of climate model, in which the different forcing will be better represented. The increase of model resolution and the additional implementations in the number of earth system components for most ESMs will provide a more realistic simulation and assessment of the impact of external forcings for sub-continental climate changes.

Volcanic eruptions are among the most prominent drivers of natural climate variability. Reconstructions for the CE show clear relationships between well-documented eruptions and climate impacts, for example the April 1815 CE Mount Tambora eruption and the subsequent "year without a summer" (Stommel and Stommel, 1983; Raible et al., 2016 for a review). In addition to short-lived effects on the radiative balance, volcanic events can have long-lasting effects. Clusters of eruptions have been proposed as the major contribution for the transition from the MCA to the LIA (Miller et al., 2012; Lehner et al., 2013), and for the long-term global cooling trend during the pre-industrial CE (McGregor et al., 2015).

Whereas model simulations generally reproduce the summer cooling, as well as aspects of regional and delayed responses to volcanic eruptions (Zanchettin et al., 2012, 2013; Atwood et al., 2016), there are discrepancies between model results and the observed climate evolution, in particular regarding the amplitude of the response to volcanic eruptions (e.g. Brohan et al., 2012; Evans et al., 2013; Wilson et al., 2016; Anchukaitis et al. 2010). Possible reasons for this disagreement include shortcomings in the volcanic reconstructions used to drive the models, or in the realism of the implementation of the aerosol forcing in the model schemes, deficiencies in reproducing the dynamic responses in the atmosphere and ocean (e.g., Charlton-Perez et al., 2013; Ding et al., 2014) or sampling biases (Anchukaitis et al., 2012; Lehner et al., 2016). The recent review by Kremser et al. (2016) concluded that the uncertainty arising from calibration of the aerosol properties to the observational period propagates into the estimated magnitude of the inferred responses in the stratospheric aerosol reconstructions. Taking into account nonlinear aerosol microphysics processes for the calculation of the volcanic aerosol radiative forcing (RF) has improved the compatibility between reconstructed and simulated climate (Timmreck et al., 2009; Stoffel et al., 2015). However, differences in the complexity and technical implementation of aerosol microphysics can lead to considerable differences in the resulting RF, even when the same sulphur dioxide injections are prescribed (Timmreck, 2012; Zanchettin et al., 2016).

Solar irradiance changes can be a significant forcing factor on decadal to centennial time scales (Gray et al., 2010). The generally cooler conditions during the LIA have often been attributed to the co-occurring grand minima, such as the

Maunder Minimum (1645-1715 CE; Eddy, 1976) that was characterized by an almost total absence of sunspots during the Maunder Minimum. However, attribution studies indicate that reduced solar forcing had a smaller impact on surface temperatures during the LIA compared to contemporary volcanic activity (Hegerl et al. 2011; Schurer et al., 2013, 2014; see also Bindoff et al., 2013).

Prior to PMIP3/CMIP5, simulations of the last millennium have used solar reconstructions with a relatively broad range of Total Solar Irradiance (TSI) variations (0.05 – 0.29%) as characterized by the change from the Late Maunder Minimum (ca. 1675 – 1715 CE, Luterbacher et al., 2001; LMM hereafter) to the late 20$^{th}$ century (1960 to 1990 CE (e.g., Ammann et al., 2007; Fernández-Donado, 2015). Note that a 0.25% change is equivalent to a variation of about 3.4 Wm$^{-2}$ in TSI. However, the higher TSI changes since the LMM, provided mostly by earlier calibrations based on the analysis of data from Sun-like

stars (Baliunas et al., 1995), were found to be unjustifiable in the light of re-analysis of stellar data by Hall and Lockwood (2004) and Wright (2004) (see also the review by Solanki et al., 2013). Therefore, the revised solar forcing reconstructions presented in Schmidt et al. (2011) exhibit typical LMM-to-present changes of 0.04 to 0.1%. Based on independent alternative assumptions for the calibration of grand solar maxima, Shapiro et al. (2011) derived a solar forcing reconstruction that exhibited a much larger long-term modulation (~0.44%) than any other. This data set was included in the update of the

PMIP3 *past1000* protocol by Schmidt et al. (2012). Later assessment of the Shapiro et al. (2011) reconstruction (Judge et al., 2012 and references therein) indicated, however, that its large amplitude is likely an overestimation (see below).

Because reconstructions of past solar forcing tend to cluster in simulations using either relatively high (i.e. mostly pre-PMIP3) or low (PMIP3) estimates of solar variations, several studies have investigated which of these provide a better fit to temperature reconstructions, but the results have so far been mixed. Whereas simulations with larger TSI variability give a

somewhat better representation of the size of the MCA – LIA transition for Northern Hemisphere temperatures (Fernández-Donado et al., 2013), statistical assessment (Hind and Moberg, 2013; Moberg et al., 2015; Pages2k-PMIP3 Group, 2015) and more detailed regional analyses (e.g., Luterbacher et al., 2016) were inconclusive. The significantly higher-amplitude reconstruction by Shapiro et al. (2011) was used in a climate model of intermediate complexity (Feulner, 2011), the HadCM3 climate model (Schurer et al., 2014), and the SOCOL model (Anet et al., 2014). Whereas the first two studies

reported a climate response incompatible with reconstructions, Anet et al. (2014) argued that high-amplitude forcing variations were necessary in their model to reproduce the cooling during the Dalton Minimum.

One of the major anthropogenic influences on the climate system over the past 2000 years was land cover change as a result of conversion of natural vegetation, mainly to agricultural and pastoral uses. The climatic effects of anthropogenic land cover change (ALCC) are undisputed in the modern world, and it is increasingly recognized that land use in the late preindustrial Holocene may have also had substantial effects on climate. In parts of the world where ALCC led to quasi-

permanent deforestation and where climate is tightly coupled to land surface conditions, we might expect regional climate to have been strongly influenced by biogeophysical feedbacks (e.g., Cook et al., 2012; Dermody et al., 2012; Pongratz et al., 2009; Strandberg et al., 2014). Additionally, permanent deforestation and loss of soil carbon as a result of cultivation (e.g., Kaplan et al., 2011; Pongratz et al., 2009) may have been substantial enough to affect global climate through the

biogeochemical feedback of $CO_2$ emissions to the atmosphere (Ruddiman et al., 2016). These effects are, however, controversial (Kaplan, 2015; Nevle et al., 2011; Pongratz et al., 2012; Stocker et al., 2014).

## 3. The Experiments

PMIP discriminates between the experiments that are endorsed by the World Climate Research Program (WCRP) CMIP6 committee (PMIP4 "tier-1": *Past1000, Mid Holocene & Last Interglacial, Last Glacial Maximum,* and *Mid Pliocene Warm*

*Period,* see Kageyama et al., 2016) and additional simulations (PMIP4 "tier-2" and "tier-3") that are more tailored to specific

interests of the palaeoclimate modelling community. This distinction is motivated by the PMIP3 experience that only a limited number of participating groups were able to afford computational resources for multiple multi-centennial simulations. In contrast to the PMIP3 protocol, PMIP4-CMIP6 recommends a single collection of external forcing data sets (the default forcing) in the "tier-1" experiments while encouraging exploration of forcing uncertainty as part of dedicated "tier-2" experiments. These "tier-2" experiments only differ in their characteristics and combination of the external drivers from the "tier-1" *past1000* experiment. The additional "tier-3" experiments are designed to allow clusters of modelling groups to perform dedicated research by exploring either specific episodes or extending them beyond the 1$^{st}$ millennium AD back in time. An overview for the experiments is given in table 1

The PMIP4-CMIP6 *past1000* simulations will build on the CMIP6 Diagnostic, Evaluation, and Characterization of Klima (DECK) experiments (Eyring et al., 2016), in particular the "pre-industrial" control (*piControl*) simulation as a reference with non-varying forcing reflecting the boundary conditions at 1850 CE. The *past1000* simulations are closely related to the CMIP6 *historical* (1850 to 2014 CE) simulations, for which they may provide more appropriate initial conditions than unforced *piControl* runs. It is expected that a number of modelling groups will be able to deliver multiple realizations of the standard *past1000* experiment.

The model versions used to carry out PMIP4-CMIP6 simulations have to be the same as those documented by the respective CMIP6 *DECK* and *historical* simulations. It is mandatory to complement the transient *past1000* and *past2k* simulations with *historical* experiments following the respective CMIP6 protocol (Eyring et al., 2016).

### 3.1 Initial state

The pre-industrial millennium is defined as covering the period 850 to 1849 CE. With the exception of the PMIP4 experiment "*past2K*" and the VolMIP-related experiment "past1000-volc-cluster" (see below), all *past1000* simulations start in 850 CE. As in PMIP3, this date was chosen in order to start the simulations significantly earlier than the MCA, which occurred at the beginning of the last millennium (ca. 950 – 1250 CE). Another reason is that the mid-to-late 9$^{th}$ century CE is estimated to have been a relatively quiet period in terms of external forcing variations or occurrence of volcanic events (e.g., Sigl et al., 2015; Bradley et al., 2016). To provide initial conditions for the simulations, it is recommended that a spin-up simulation is performed departing from the CMIP6 *piControl* experiment with all forcing parameters set to ~850 CE values. The length of this spin-up simulation will be model- and resource- dependent. However, it should be long enough to minimize at least surface climate trends (Gregory, 2010). The spin-up has to be documented and this should include information on a few key variables (see section 3.6). The spin-up should be consistent with the *piControl* (for example, it should include a background volcanic aerosol level, and appropriate anthropogenic modifications to land use/land cover characteristics (as for the *piControl* simulation; see Eyring et al., 2016).

### 3.2 PMIP4-CMIP6 Tier1: The standard PMIP4-CMIP6 *past1000* simulation plus CMIP6 *historical* simulation

The standard PMIP4-CMIP6 *past1000* experiment applies the default forcing data set (see below) and is complemented by an *historical* (1850 – 2014 CE) simulation that uses the end state of the *past1000* simulation in 1850 CE for initialization and that follows the CMIP6 protocol (Eyring et al., 2016). This procedure provides a consistent data set for past and present climate variations. Comparing historical simulations initialized from a *piControl* run (the CMIP6 default) with those starting from 1849 CE conditions from *past1000* serves to assess the impact of initial conditions on the evolution of the 19$^{th}$ and 20$^{th}$ century climate.

Modelling groups are encouraged to extend this set of experiments to multiple realisations, using the same forcing, but with perturbed initial conditions. While an ensemble size of ten has been shown to be desirable (Otto-Bliesner et al., 2016;

Stevenson et al., 2016), we acknowledge that limitations in computational resources or high computational demand of high-resolution models may prevent groups from producing large ensembles.

### 3.3 PMIP4 Tier-2: Forcing Uncertainty and Attribution

The "tier-2" category experiments are recommended to further explore uncertainties related to external drivers. Without taking uncertainties in forcing into account, model/observation discrepancies might be wrongly attributed to model failures and/or systematic problems in proxy reconstructions. The "tier-2" *past1000* experiments should be set-up in a similar way as the "tier-1" *past1000* experiment, i.e. the simulation should cover the period 850 to 1849 CE and the same initial conditions should be used. As for "tier-1", there should be a *historical* simulation complementing each "tier-2" *past1000* simulation. For experiment naming and identification, see table 1.

### 3.3.1 Alternative forcings:

 Uncertainties in the reconstruction of forcing agents are associated with the source data (mostly proxies), reconstruction methodology, calibration to records representing present conditions, or with the way that the forcing time series are deduced from more explicit modelling approaches. PMIP4 provides forcing data sets derived through different methodologies (e.g., for solar irradiance, see below), as well as different versions of the same forcing data set (e.g., by varying parameters in the construction scheme). It also promotes the assessment of independently derived reconstructions that will become available during the evolution of PMIP4. For example, modelling groups are encouraged to explore and document the impact on simulated climate resulting from variations in volcanic forcing associated with the uncertainty in the translation from sulphur injections to aerosol optical properties.

### 3.3.2 Individual forcing agents

 The role of individual drivers can be assessed by performing single-forcing simulations (e.g., Pongratz et al., 2009; Schurer et al., 2014; Otto-Bliesner et al., 2016). However, low signal-to-noise ratios and the dependence of the response to varying background conditions (Zanchettin et al., 2013) require careful analyses and will be most beneficial if performed in ensemble mode (Schurer et al., 2014; Otto-Bliesner et al., 2016).

### 3.4 PMIP4 Tier-3: Additional experiments

The "tier-3" category experiments will enable clusters of modelling groups to perform dedicated research by exploring either specific episodes or advancing the scope of the *past1000* simulations. For experiment naming and identification, see table 1.

### 3.4.1 Volcanic forcing and climate change in the early instrumental period:  the *past1000-volc-cluster*

Because many groups will not be able to perform ensemble simulations over the entire period, we suggest performing multiple realisations of the early 19[th] century. This period is characterized by relatively strong variations in solar activity, including the Dalton Minimum, and strong volcanic eruptions in 1809, 1815, and 1835 CE. It is the coldest period of the past 500 years, and it is well documented as part of the early instrumental period (e.g. Brohan et al., 2012). The experiment will be carried out in cooperation with the Model Intercomparison Project on the climatic response to Volcanic forcing (VolMIP, Zanchettin et al., 2016). The experiment requires an ensemble (minimum three members) of 70-year long simulations starting from *past1000* restart files in 1790 CE. In contrast to the VolMIP experiment "*volc-cluster-mill*", all external drivers remain active.

### 3.4.2 The *past2K* experiment

 With the advent of longer reconstructions, in particular for volcanic eruptions (e.g., Sigl et al., 2015; Toohey and Sigl, 2017), it is now possible to start the simulations at the beginning of the 1[st] millennium CE. In fact, except for the land-use change forcing, all forcing reconstructions described above for the "tier-1" *past1000* experiment are available for the entire

CE and the groups need to make sure that the same forcing is used for *past1000* and *past2k* during the period 850 to 1849 CE. Additional forcing reconstructions (e.g., land-use) will be completed during the course of PMIP4. The *past2k* simulations will provide a basis for the analyses of specific periods in the 1$^{st}$ millennium CE that have attracted attention based on historical evidence, for instance those related to the Roman Empire (Büntgen et al., 2011; Luterbacher et al., 2016) and to the onset and evolution of the "Late Antique Little Ice Age" (Büntgen et al., 2016; Toohey et al., 2016). Additionally, there is a growing archive of lower resolution syntheses of marine sediment-based reconstructions that span the full CE (Marcott et al 2013; McGregor et al 2015). The *past2K* experiment will allow the community to better investigate the full span of the Medieval period and its temporal evolution, as the start of the *past1000* experiment in the year 850 CE might neglect some important initial conditions constrained during preceding periods (see also Bradley et al., 2016). Prior to the start of the experiment, a spin-up procedure similar to the *past1000* experiment has to be undertaken for year 1 CE conditions.

### 3.4.3 Including an interactive carbon cycle: the *past1000esm* experiment

PMIP4 will extend the scope of the *past1000* experiment and include simulations with models that include an interactive carbon cycle. Complementing the experiments *esm-piControl* and *esm-hist* performed by the Coupled Climate Carbon Cycle Modelling Intercomparison Project (C4MIP; Jones et al., 2016), carbon cycle feedbacks and interaction will be studied in the pre-industrial millennium.

### 3.5 Experiment identification

The experiments are defined by their short name (e.g., *past1000*) and an extension following the "ripf" classification, where "r" stands for "realization, "i" for initialization, "p" for perturbed physics, and "f" for forcing (Table 1). The letters r, i, p, and f are followed by integers K, L, M, and N, respectively. For example, different realisations within an ensemble would have different values for "K" following the "r". To classify a simulation with a model with modified physical parameterization, one would vary the integer "M" after the "p". The experiments using the default forcing are defined by "f1", alternative or single forcing would be identified by a different integer value "N". CMIP6 *historical* simulations starting from a *past1000* run should vary the integer "L" after the "i".

### 3.6 Documenting the simulations

The modelling groups are responsible for a comprehensive documentation of the model system and the experiments. A PMIP4 special issue in GMD and Climate of the Past has been opened where the groups are encouraged to publish these documentations. The documentation should include:

- The model version and specifications, like interactive vegetation or interactive aerosol modules etc.
- A link to the DECK experiments performed with this model version
- Specification of the forcing data sets used and their implementation in the model
- A documentation of the spin-up strategy to arrive at 850 CE (1 CE for past2k) initial conditions. We request providing information on drift in key variables for a few hundred years at the end of the spin-up and the beginning of the actual experiment. These variables are:
  - globally and annually averaged SSTs
  - deep ocean temperatures (global and annual average over depths below 2500m)
  - deep ocean salinity (global and annual average over depths below 2500m)
  - top of atmosphere energy budget (global and annual average)

- surface energy budget (global and annual average)

- northern sea-ice (annual average over northern hemisphere)

- southern sea-ice (annual average over southern hemisphere)

- northern surface air temperature (annual average over northern hemisphere)

- southern surface air temperature (annual average over southern hemisphere)

- Atlantic Meridional Overturning Circulation (maximum overturning in the North Atlantic basin)

- carbon budget by the biosphere.

### 3.7 Output variables and data distribution

The "tier-1" *past1000* simulation is part of the CMIP6 experiment family and output data will be distributed through the official CMIP6 channels via the Earth System Grid Federation (ESGF,

https://earthsystemcog.org/projects/wip/CMIP6DataRequest).

Data from PMIP4-only "tier-2" and "tier-3" simulations must be processed following the same standards for data processing (e.g. CMOR standards) and should be distributed via ESGF.

Groups contributing *past1000* simulations to CMIP6-PMIP4 should ideally deliver the entire set defined in the data request. However, an important issue for long-term simulations such as *past1000* is storage demand for high-frequency output. As a minimum, we ask for a subset of two-dimensional daily variables that allow investigations on extreme events and particular dynamical features, including near surface air temperature (tas), daily maximum near surface air temperature (tasmax), daily minimum near surface air temperature (tasmin), daily maximum near-surface wind speed (sfcWindmax), precipitation (pr), sea-level pressure (mslp), 500 hPa geopotential (zg500), daily maximum hourly precipitation Rate (prhmax). If storage of high-frequency output for the entire millennium should be too demanding, we recommend to concentrate efforts to three multi-decadal periods (in descending priority): 1. The early 19[th] century (1790 to 1849 CE as focus period of VolMIP), and 2. the Maunder Minimum (1645 to 1715 CE) and 3.) the Medieval Climate Anomaly (1100 to 1170 CE) covering periods of high and low solar activity, respectively.

Groups participating in PMIP and VolMIP should pay attention to the new diagnostics of volcanic instantaneous radiative forcing defined by VolMIP, whose calculation is recommended for some major volcanic events simulated in the *past1000* experiment (for details, see Zanchettin et al., 2016). Groups that run the PMIP4-CMIP6 experiments with the carbon cycle enabled should pay attention to the output variables requested by OCMIP and C4MIP.

The list of variables requested by PMIP for the PMIP4-CMIP6 palaeoclimate experiments can be found here:

http://clipc-services.ceda.ac.uk/dreq/u/PMIP.html. This request is presently processed by the CMIP6 Working Group for Coupled Modeling Infrastructure Panel (WIP) into tables, which define the variables included in the data request to the modelling groups for data to be contributed to the archive. The most up-to-date list including all variables requested for CMIP6 can be found at the WIP site:

proj.badc.rl.ac.uk/svn/exarch/CMIP6dreq/tags/latest/dreqPy/docs/CMIP6_MIP_tables.xlsx

The last two columns in each row list MIPs associated with each variable. The first column in this pair lists the MIPs, which are requesting the variable in one or more experiments. The second column lists the MIPs proposing experiments in which this variable is requested.

As supplementary to this manuscript we provide version 1.00.12 (June 2017) of the table. We note, however, that this document is still in development and inconsistencies may still exist.

## 4. Description of forcing boundary conditions

Some of the forcing fields are extensions in time of the "official" CMIP6 data sets for the *historical* simulations. These are documented in individual contributions to the GMD special issue on CMIP6 and available through the contributors' web sites (see below and Appendices). PMIP4 specific time series and reconstructions are available via the PMIP4 website and specifications on data format and technical implementation are given in the Appendices.

### 4.1 Orbital forcing

Over the pre-industrial millennium, the orbital forcing is dominated by changes in the perihelion, whereas variations in eccentricity and obliquity are rather small (Berger, 1978; see also Figure 1 in Schmidt et al., 2011). The orbital forcing remains unchanged from what was used in PMIP3 (Schmidt et al., 2011). Note, however, that the reference insolation year is 1860 CE in CMIP6 (Eyring et al., 2016), compared to 1950 in PMIP3. Unless the models calculate the orbital parameters internally, groups will use a list of annually varying orbital parameters (eccentricity, obliquity, and perihelion longitude), changing every January 1st (see Appendix A1).

### 4.2 Greenhouse gas forcing

GHG time-series for concentration-driven simulations are provided by CMIP6 for the period 1 CE to 2014 CE (Figure 1). The data compilations for surface concentrations of $CO_2$, $CH_4$, $N_2O$ are based on updated instrumental data and ice-core records (Meinshausen et al., 2017). Differences between the new CMIP6 data set and previous estimates for CMIP5 are rather small (e.g., for global mean surface mixing rations see figure 9 in Meinshausen et al., 2017). The CMIP6 reconstruction offers better representation of latitudinal and seasonal variations and we recommend using this data set for consistency throughout the CE. GHGs should be implemented as for the CMIP6 historical simulations (see http://www.climatecollege.unimelb.edu.au/cmip6 and Appendix A2).

### 4.3. Volcanic forcing

Based on newly compiled, synchronized and re-dated high-resolution, multi-parameter records from Greenland and Antarctica (Sigl et al., 2014, 2015), the eVolv2k time series of volcanic stratospheric sulphur injections has been developed by Toohey and Sigl (2017). Discrepancies in the timing of volcanic events recorded in ice cores and short-term cooling events in proxy-based temperature records have been largely resolved by improvements in absolute dating of the ice core record (Sigl et al., 2015). This was based on the detection of an abrupt enrichment event in the [14]C content of tree rings (Miyake et al., 2012) and the tuning of the ice core chronology based on matching the corresponding [10]Be peak (Sigl et al., 2015). The Toohey and Sigl (2017) data set is the recommended forcing for the PMIP4-CMIP6 *past1000* experiments (see Appendix A3). Modelling groups using interactive aerosol modules and sulphur dioxide injections in their *historical* simulations follow the same method for the *past1000* experiment and can use the sulphur dioxide injection estimates directly. For other models, aerosol radiative properties as a function of latitude, height, and wavelength can be derived by means of the Easy Volcanic Aerosol (EVA) module (Toohey et al., 2016). EVA uses the sulphur dioxide injection time series as input and applies a parameterized three-box model of stratospheric transport to reconstruct the space-time structure of sulphate aerosol evolution. As outlined in more detail in Toohey et al. (2016), simple scaling relationships serve to construct mid-visible aerosol optical depth (AOD) and aerosol effective radius ($r_{eff}$) from stratospheric sulphate aerosol mass. Finally, wavelength dependent aerosol extinction, single scattering albedo and scattering asymmetry factors are

derived for user-defined latitude and wavelength grids. Volcanic forcing files produced with EVA have the same fields and format as the recommended volcanic forcing files for the CMIP6 historical experiment (see https://www.wcrp-climate.org/wgcm-cmip/wgcm-cmip6) and allow for consistent implementation in different models.

Global mean AOD time-series produced by EVA using the eVolv2k sulphur dioxide injection time series show relatively good agreement with the previous PMIP3 reconstructions over the past 1000 years, although some important differences exist. Figure 2 shows the 850-1850 CE time series of global mean mid-visible (550 nm) AOD produced by EVA using the eVolv2k sulphur injection time series (hereafter EVA2k) compared to the forcing reconstructions by Gao et al., (2008; hereafter denoted as GRA08) and Crowley and Unterman (2013; hereafter CU13). Note that the sulphate aerosol mass provided by the GRA08 reconstruction has been converted here to AOD by assuming a constant scaling factor as in Schmidt et al. (2011), although this may not reflect the actual radiative impact attained with different methods of implementation used in different climate models. The largest discrepancy between the GRA08 and CU13 reconstructions was the magnitude of forcing associated with the 1257 CE Samalas eruption, with GRA08 prescribing a forcing about twice as large as that of CU13. The magnitude of the Samalas forcing in the EVA2k reconstruction is more similar to that of CU13. In the late 18[th] century, the EVA2k forcing is stronger than that of CU13, and more consistent with the GRA08 reconstruction, because the CU13 reconstruction included a correction to the ice core sulphate signal of the 1783 CE Laki eruption. The forcing for this eruption therefore could be overestimated in EVA2k and GRA08 if the ice core record represents mostly sulphate of tropospheric rather than stratospheric origin. The EVA2k and GRA08 reconstructions are also stronger than CU13 in the late 12[th] century, due to the identification of a series of large eruptions during this period. Prior to around 1150 CE, the EVA2k reconstruction shows little correlation with the other reconstructions, due to a change in the ice core age-model (Sigl et al., 2015) and identification of additional volcanic events (Sigl et al., 2014). This period is characterized by less frequent and less intense volcanic activity compared to earlier and subsequent periods, although the difference between this "quiet" period and periods of strong activity is somewhat smaller in EVA2k compared to the other forcing reconstructions. An important difference compared to previous forcing data sets is that the new EVA2k reconstruction includes a background stratospheric aerosol level, which produces a non-zero minimum AOD in periods of no volcanic eruptions. Like the CMIP6 historical volcanic forcing, the background level is defined to be equal in global mean AOD to the observed AOD minimum in the years 1999-2000 CE (Thomason et al., 2016).

The reconstruction of volcanic forcing from ice core records carries substantial uncertainties (Hegerl et al., 2006; Gao et al., 2008; Crowley and Unterman, 2013; Stoffel et al., 2015). At present, different global aerosol models produce a large range of forcing estimates for specified sulphur injections, which motivates on-going research (Zanchettin et al., 2016). The EVA module allows for the production of volcanic forcing time series with varying characteristics, such as the magnitude of the eruptions. By modifying an internal parameter, which converts stratospheric sulphate mass to aerosol optical depth, the magnitude can easily be adjusted. Variations in this parameter can be used to reflect the overall systematic uncertainty in the estimation of the volcanic forcing. Alternative volcanic forcing time-series deduced from global aerosol models will provide further volcanic forcing options for dedicated experiments.

### 4.4 Solar variations

The reconstruction of solar activity before the telescope-era (i.e. before 1610 CE) relies on records of cosmogenic isotopes such as $^{14}$C or $^{10}$Be. Both radionuclides are produced in the terrestrial atmosphere by cosmic rays and their production is modulated by solar activity and the geomagnetic field. After production, they take different pathways and are influenced by different environmental conditions before their deposition in terrestrial archives (e.g., McHargue and Damon, 1991; Beer et al., 2012). Despite some discrepancy between $^{10}$Be and $^{14}$C-based reconstructions on decadal and sub-decadal time scales, they agree well on the centennial-millennial time scales (Bard et al., 2000; Vonmoos et al., 2006; Usoskin et al., 2009;

Steinhilber et al., 2012). PMIP4 provides new reconstructions of TSI and Spectral Solar Irradiance (SSI) that are based on recent reconstructions of cosmogenic isotope data [14]C (Roth and Joos, 2013; Usoskin et al., 2016b) and [10]Be (Baroni et al., 2015). Solar surface magnetic flux and the equivalent sunspot numbers are reconstructed from the isotope data through a chain of physics-based models (see Appendix A4 and Vieira et al., 2011; Usoskin et al., 2014, 2016b). Because only decadal

values of the sunspot number and the open magnetic flux can be reconstructed in this way, the 11-year solar cycle has to be reconstructed separately. This is done employing statistical relationships relating various properties of the solar cycle derived from direct sunspot observations (Wu et al., in prep.).

The reconstructed yearly sunspot number is then fed into irradiance models, to produce TSI and SSI records. We employ two different models, namely the updated SATIRE-M model (Vieira et al. 2011; Wu et al., in prep.) and an update of the Shapiro

et al. (2011) model (PMOD hereafter, reflecting its origin from the Physikalisch-Meteorologisches Observatorium Davos). For the SATIRE-based reconstructions, the amplitude of the variations on time scales of centuries is comparable in magnitude with the PMIP3 reconstruction by Vieira et al. (2011). In response to the findings of Judge et al. (2012), the PMOD model is revised such that the long-term change in the quiet Sun is interpolated between the models "B" and "C" of Fontenla et al. (1999), instead of the "A" and "C" models. This reduces the recovered secular change in TSI between the

Maunder minimum and the present by almost a factor of two (Egorova et al., 2017). Nevertheless, the centennial variations are still much larger than in the SATIRE-based data sets (Figure 3). As pointed out by Schmidt et al. (2012), the uncertainty in the PMOD reconstruction is relatively high and this forcing should be considered as an upper limit of the possible secular variability. For the PMOD reconstruction, only a [14]C-based version is provided.

Both irradiance models employ semi-empirical model atmospheres to describe the brightness spectra of the various solar

surface components (sunspots, faculae, network) responsible for solar irradiance variability on time scales of days to millennia. This allows the consistent reconstruction of both TSI and SSI without relying on SSI measurements. The reconstructions agree with measurements in periods, where the latter are considered reliable (cf. Ermolli et al. 2013; Yeo et al. 2015). All provided reconstructions are normalised to give the revised absolute TSI level of 1361 W/m$^2$ during the most recent activity minimum in 2008, as measured by SORCE/TIM (Kopp, 2014). Differences in the secular variations in TSI

(Figure 3) are mainly due to the assumptions made in the irradiance models. The new PMOD-based reconstruction features a LMM-to-present amplitude of 3.4 Wm$^{-2}$ (about 0.25%) whereas the SATIRE-based forcing changes by less than 1 Wm$^{-2}$ (0.06%) during this period. Differences between the [14]C and [10]Be based reconstructions manifest themselves mainly in the phasing and differences in secular trends, for example in the duration and timing of the LMM. The SATIRE-based solar activity reconstruction is also in good overall agreement with a solar activity reconstruction that is exclusively based on

cosmic ray measurement/proxy data via a combination of [14]C and neutron monitor data (Muscheler et al., 2016).

To achieve a smooth transition from the pre-industrial to the modern period, the reconstructions are combined (see Appendix A4.2 for details) with the solar forcing records recommended for the CMIP6 *historical* experiment (Matthes et al., 2017). This transition is essentially straightforward for TSI. However, some artefacts cannot be avoided for SSI. The CMIP6 *historical* solar forcing is derived from an average of two conceptually different models, NRLSSI-2 (Coddington et al. 2015)

and SATIRE, where the latter is a splice of SATIRE-T, based on sunspot observations before 1874 CE (Krivova et al., 2010) and SATIRE-S, based on solar full-disc magnetograms afterwards (Yeo et al., 2014). Differences between the NRLSSI and SATIRE models are discussed by Yeo et al. (2015). Averaging the two intrinsically different SSI series yields a record in which the shape of the solar spectrum does not conform to either model or to observations, e.g., the ATLAS3 (Thuillier et al., 2003) or WHI (Woods et al., 2009) quiet Sun reference spectra.

The SSI records provided for the PMIP4 experiments are a combination of the rescaled reconstructions before 1850 CE, shown for the [14]C-based SATIRE reconstruction data set as the cyan solid line in Figure 4, and the CMIP6 time series for the *historical* simulations (Matthes et al., 2017), shown by the red line. Details of the rescaling and adjustment can be found in

Appendix 4.2. Compared to the original reconstruction, the CMIP6 record underestimates the variability in the UV after 1850 CE by about 10-15%, and by more than 35% if compared to PMOD (not shown), while it overestimates the variability in the visible and IR by about 10-15% and by more than 40%, respectively. While adjusting the pre-industrial reconstruction to the CMIP6 historical records yields a smooth transition in 1850 CE, it needs to be kept in mind that the amplitude of the variability in the spectral bands is adopted from the original models (i.e. from isotope-based reconstructions before 1850 CE and the CMIP6 record afterwards) and depends at least partly on the construction of the dataset. In addition to the standard (adjusted to CMIP6) [14]C data sets, we therefore also provide the original records for the entire period for testing the climatic effects of the conflation.

In summary, PMIP4 provides three reconstructions of TSI and SSI from the most-up-to-date records of cosmogenic radioisotopes [14]C and [10]Be using a chain of models, all of which have been improved and updated since PMIP3. In contrast to CMIP3, for all provided reconstructions, total and spectral irradiance are computed in a self-consistent manner. In particular, the same model has been used to reconstruct irradiance from each radioisotope to allow an estimate of the uncertainty due to the effect of local conditions on their formation and deposition. Two irradiance reconstructions were obtained from [14]C data using different irradiance models to allow for sensitivity experiments testing the response to the amplitude of the solar forcing. The default forcing for CMIP6-PMIP4 *past1000* is the [14]C SATIRE-based reconstruction. The PMOD-based reconstruction provides an upper limit on the magnitude of the long-term changes in irradiance. Since the historical CMIP6 recommendation is an arithmetic average of two conceptually different models with significant differences in the SSI variability, special care has been taken to combine the PMIP4 data sets with the historical forcing. The approach we have chosen here allows for a smooth transition but might nevertheless produce some artefacts.

Apart from the direct effect of changes in TSI and SSI, solar variability also affects stratospheric and mesospheric ozone abundances (e.g. Haigh, 1994) and can contribute significantly to the total stratospheric heating response. In climate models including interactive chemistry the photolysis scheme should adequately simulate the ozone response to variations in the UV part of SSI. CMIP6 models that do not include interactive chemistry should prescribe ozone variations consistent with the solar forcing and apply a scaling approach similar to the one recommended for the historical period (Matthes et al., 2017; Maycock et al., 2016b). It should be noted that solar-ozone regression coefficients as provided by Maycock (2016b) have been calculated with respect to the 10.7cm radio flux (F10.7), which is not available for the PMIP period. Hence we have re-performed the regression of the same ozone fields but with respect to solar UV irradiance averaged over the spectral range from 200 to 320 nm (see Appendix 4.3 for details). We recommend calculating time varying ozone input for PMIP4 by scaling these coefficients with the anomaly of the respective UV flux during the simulation period and add it to the CMIP6 preindustrial ozone climatology. The UV flux anomaly should accordingly be calculated with respect to the CMIP6 preindustrial irradiance data (Matthes et al., 2017).

### 4.5 Land use changes and anthropogenic land cover changes

For the *past1000* simulation, land-use changes need to be implemented using the same input datasets and methodology as the historical simulations; the CMIP6 land-use forcing datasets now cover the entire period 850-2015 CE (Hurtt et al., in prep.), which provides a seamless transition between the CMIP6 *past1000* and *historical* simulations. The new land-use forcing, Land-Use Harmonization 2 (LUH2), is provided as a contribution of the Land-Use Model Intercomparison Project (LUMIP) to CMIP6 (https://cmip.ucar.edu/lumip). The LUH2 strategy estimates the fractional land-use patterns, underlying land-use transitions, and key agricultural management information, annually for the period 850-2100 CE at 0.25° x 0.25° spatial resolution. The estimate minimizes the differences at the transition between the historical reconstruction and the conditions derived from Integrated Assessment Models (IAM). It is based on new estimates of gridded cropland, grazing lands, urban land, and irrigated land, from the Historical Land Use Data Set for the Holocene (HYDE3.2, Klein Goldewijk, 2016). Within

HYDE3.2, grazing lands are now sub-divided into managed pasture and rangeland categories, and irrigated land also includes a sub-category of land flooded for paddy rice. Within LUH2, cropland area is sub-divided into five crop functional types based on data from Monfreda et al. (2008) and from the Food and Agricultural Organisation of the United Nations (FAO). The temporal evolution of the various types is displayed in Figure 5. LUH2 includes a new representation of shifting cultivation rates and patterns and also includes new layers of management information such as irrigated area and industrial fertilizer usage.

As wood was the primary fuel and an important construction material for nearly all societies in the preindustrial world, LUH2 includes new scenario reconstructions of wood consumption for the period 850 to 2014 CE. To build these scenarios, an estimate of a baseline wood demand following McGrath et al. (2015) was compiled. To account for differences between continents and technology-induced changes in consumption patterns over time, the wood demand was scaled by historical, country-level estimates of Gross Domestic Product (GDP) (Maddison, 2003; Bolt and van Zanden, 2014) (see appendix A5.1 for details).

As in PMIP3/CMIP5, the default land use dataset is at the lower end of the spread in estimates of early agricultural area indicated by other reconstructions (Pongratz et al., 2008; Kaplan et al., 2011). In turn, the lower estimate of early agricultural area at the beginning of the last millennium implies larger land-use-induced land cover changes over time to match the land cover distribution of the industrial era (see Schmidt et al., 2012). To allow an assessment of the substantial uncertainties associated with reconstructing historical land use, while at the same time remaining consistent with the format of the default dataset, maximum and minimum alternative reconstructions of the LUH2 dataset will also be provided during the course of PMIP4. In particular, both upper and lower-bound scenarios will be created in order to provide a range of wood consumption scenarios. The upper scenario is identical to the baseline scenario but without the national scale factors based on Smil (2010). The lower scenario uses the 1920 CE per capita rates from Zon and Sparhawk (1923) for all years prior to 1920 CE.

Note that because most of the PMIP4 simulations are driven by prescribed GHG concentrations, the effect of land use change on atmospheric GHG composition is captured by the GHG forcing. The land use forcing thus does not affect the atmospheric $CO_2$ concentration, although the terrestrial carbon cycle will be substantially affected. Combined land use and fossil-fuel-related carbon fluxes can be diagnosed as implied emissions (e.g., Roeckner et al., 2010). Nevertheless, the key climate effects from the land use forcing in the concentration-driven setup stems from the biogeophysical effects, i.e. changes in energy and water balance due to altered land surface characteristics, which alter climate in particular at the regional level (e.g., Brovkin et al., 2013).

## 5. Role of *past1000* simulations in CMIP and links to WCRP "Grand Challenges"

Simulations of the last millennium directly address the first CMIP6 key scientific question "How does the Earth System respond to forcing?". Investigating the response to (mainly) natural forcing under climatic background conditions that are not too different from today is crucial for an improved understanding of climate variability, circulation, and regional connectivity. In providing in-depth model evaluation with respect to observations and palaeo-climatic reconstructions, and specifically by comparing details of the simulated response to forcing to that of observations, *past1000* simulations serve to "understand origins and consequences of systematic model biases". Furthermore, they allow the assessment of observed and simulated climate variability on decadal to centennial time scales, and provide information on predictability under forced and unforced conditions. These are important elements for making near-term predictions and for providing robust attributions of past change and thus address the third CMIP6 scientific question "How can we assess future climate changes given climate variability, predictability and uncertainties in scenarios?"

The *past1000* simulations focus on the assessment of forced vs. internal variability and provide context for present and future changes. Research stimulated by PMIP will therefore link to the "Grand Challenges" of the WCRP (Brasseur and Carlson, 2015). In particular, the *past1000* simulation will contribute to the science challenges "Clouds, Circulation, and Climate Sensitivity", "Understanding and Predicting Weather and Climate Extremes", and "Carbon feedbacks in the climate system". The PMIP simulations will also provide a palaeo perspective for more impact-related themes such as "Changes in Water Availability" and "Regional Sea-level Change & Coastal Impacts".

## 5.1 Interaction with other CMIP6 MIPs and PAGES

Cooperation between PMIP and other MIPs will create synergies for climate model evaluation and improved process understanding. The *past1000* simulations provide long-term perspective on climate variability and allow for the assessment of the response to forcing for a time-period that is well constrained by reconstructions and early observations. This is particularly relevant for the Detection and Attribution MIP (Gillett et al., 2016). Changes in land-use are an important forcing factor and PMIP will benefit from research and forcing reconstructions produced in the framework of the Land-Use Model Intercomparison Project (Lawrence et al., 2016; Hurtt et al, in prep.). Together with VolMIP (Zanchettin et al., 2016), PMIP assesses different aspects of the climatic response to volcanic forcing. Whereas VolMIP focuses on idealized volcanic perturbation experiments with well-constrained forcing across participating models and well-defined initial conditions, *past1000* simulations describe the climate response to volcanic forcing in long transient simulations, where related uncertainties are partly due to chosen input data for volcanic forcing. In cooperation with VolMIP, PMIP targets the early instrumental period at the beginning of the 19th century.

PMIP will provide input to and benefit from diagnostic projects performed within the framework of the Ocean Model Intercomparison Project (OMIP, Griffies et al., 2016) and its biogeochemical component (OCMIP, Orr et al., 2016), the Sea-Ice MIP (SIMIP, Notz et al., 2016), the Flux-anomaly-forced MIP (FAFMIP, Gregory et al., 2016), and the Coupled Climate - Carbon Cycle MIP (C4MIP, Jones et al., 2016).

The PMIP Past2K working group will continue to interact with the PAGES 2k Initiative (http://www.pages-igbp.org/ini/wg/2k-network/intro) and further explore continental and regional scale features of climate change during the CE. Following the research agenda of the second phase of PAGES 2K, the focus will shift from continental-scale temperature reconstruction to understanding mechanisms of climate variability, teleconnections, spatial-temporal ocean and atmosphere dynamics and the hydrological cycle. We also envision a closer link to the PAGES Ocean2k working group investigating ocean circulation (gyre, overturning circulation, heat content changes, heat transports).

Hydroclimate is an increasing focus of the PAGES 2k proxy communities (e.g., Cook et al., 2015; Ljungqvist et al., 2016). The PMIP4-CMIP6 multi-model ensemble of *past1000* simulations allows the community to explore how climate models simulate hydroclimate change and variability, and whether they do so in ways that are consistent with the palaeoclimatic records. Such comparative analyses emphasize the methods appropriate for data-model comparisons that target hydroclimate in order to understand climate change at regional scales and the mechanisms of climate variability at decadal to centennial timescales (e.g. Coats et al., 2015b).

PMIP has provided to CMIP6 a comprehensive list of output variables that includes all necessary variables for analyses of atmospheric, oceanic and land- surface processes (see section 3.6). CMIP6 will make sure that all groups store the output variables in a consistent way (see https://earthsystemcog.org/projects/wip/CMIP6DataRequest).

By analysis of the *past1000* simulations and proxy-based reconstructions, model-data comparison exercises can help to identify mechanisms of climate variability that are not realistically simulated by present AOGCMs (e.g., the Atlantic Multidecadal Variability; Kavvada et al., 2013). Detection and attribution studies using state-of-the-art climate models will focus on attributing regional variations across the last one or two millennia, and determining the roles of GHG fluctuations,

solar variability, volcanic forcing as well as land use changes in explaining anomalies of the past. Such investigations would also benefit from the "tier-2" single-forcing simulations outlined in section 3.2.2. On the longer time horizon, new models and updated forcing, in conjunction with new reconstructions of climate variables and the ability to simulate proxies directly, will reduce uncertainty and determine model-data consistency.

## 6. Conclusions

The PMIP4-CMIP6 *past1000* simulations provide a framework for integrated studies of climate evolution during the pre-industrial period. Together with the additional *historical* simulations that are initialized from the *past1000s* in 1850 CE, they allow the community to study the transition from conditions influenced mainly by natural forcing to those determined largely by anthropogenic drivers. Improvements in PMIP4/CMIP6 relative to PMIP3/CMIP5 are expected due to new and more comprehensive reconstructions of external forcing, improved models, and improved experimental protocols that ensure seamless simulations from the pre-industrial past to the future. New, high-resolution simulations may improve the assessment of smaller-scale regional details and processes, e.g. storm-tracks or precipitation, and modes of variability. Multiple realisations will be available for a larger subset of models, enabling improved assessments of the relative contributions of internal climate variability and externally forced changes towards the evolution of the climate system over the last millennium.

The wealth of proxy-based reconstructions together with the multi-model, multi-realisation data base provided by PMIP4 simulations, will refine investigations of the response to external forcing, allow studies of regional versus global changes, and improve process understanding. Dedicated sensitivity studies will, in addition to the default *past1000* simulation, allow individual groups or clusters of researchers to investigate uncertainty in reconstructions and the representation of the forcing agents in the models. In particular, a broader evaluation of the PMIP4 simulations of the last millennium is expected due to the increasing attention on processes and variables other than temperature, such as the hydrological cycle and climate extremes. PMIP4 collaborates with other MIPs, particularly with those working on climate system mechanisms, such as VolMIP, and provides input to other MIPs that will evaluate long-term integrations (e.g., DAMIP). PMIP as an organizational body will coordinate research activities within its working groups and continue the fruitful liaison with the PAGES 2k community.

## 7. Data availability

All forcing data sets and the EVA tool for producing aerosol optical properties can be accessed via the PMIP4 *past1000* web page: https://pmip4.lsce.ipsl.fr/doku.php/exp_design:lm. The forcing data sets provided exclusively for the *past1000* simulations (orbital, solar, volcanic), can be downloaded directly from the PMIP4 repository. They will be accessible also as contribution to Input4MIPs (https://esgf-node.llnl.gov/projects/input4mips/, see the living document "Input4MIPs summary"). The CMIP6 historical forcing data sets that provide extensions into the Common Era (GHG, land-use) and that are documented in individual contributions to the CMIP6 GMD special issue are already accessible through ESGF (https://esgf-node.llnl.gov/search/input4mips/) or via links to the originators' web pages (see "Input4MIPs summary"). The results of the experiments described here will be distributed via the Earth System Grid Federation (ESGF, https://earthsystemcog.org/projects/wip/CMIP6DataRequest), as described together with the requested output variables in section 3.7.

**Authors' contribution:** J.H. Jungclaus, P. Braconnot, J. Cao, M. Evans, J.F. Gonzalez-Rouco, N. Krivova, J. Luterbacher, B. Otto-Bliesner, S.J. Phipps, G.A. Schmidt, J. Smerdon, M. Toohey, S. Wagner, and D. Zanchettin have designed the article, coordinated the writing and drafted the manuscript. The solar forcing reconstructions were assembled and described by E. Bard, M. Baroni, T. Egorova, N. Krivova, R. Muscheler, E. Rozanov, H. Schmidt, W. Schmutz, A.I. Shapiro, S.K. Solanki, I.G. Usoskin, C.-J. Wu, K.L. Yeo. H. Schmidt and A.C. Maycock provided the section on solar-related ozone variations. M. Khodri, A.N. LeGrande, S.J. Lorenz, M. Sigl, C. Timmreck, and M. Toohey provided the volcanic forcing and the description of its implementation. L.P. Chini, G.C. Hurtt, J.O. Kaplan, K. Klein Goldewijk, and J. Pongratz contributed the land-use data set and its documentation. F. Joos, M. Meinshausen, and C. Nehrbass-Ahles provided the section on greenhouse gas forcing. J. Cao, M. Evans, H. Goosse, J. Luterbacher, W. Man, A. Schurer, A. Moberg, Q. Zhang, and E. Zorita contributed to the section on specific analyses, the sections on the links between PMIP and other MIPs. All authors have contributed to the revisions and have approved of the final version of the manuscript.

**Acknowledgements:** The work by I.G. Usoskin was partly done in the framework the Center of Excellence ReSoLVE (project No. 272157 of the Academy of Finland). B. Otto-Bliesner acknowledges funding by the U.S. National Science Foundation (US-NSF) of the National Center for Atmospheric Research and support by US-NSF EaSM2 award AGS 1243107. J. Pongratz is supported by the German Research Foundation's Emmy Noether Program (PO 1751/1-1). E. Rozanov and T. Egorova have been partially supported by the Swiss National Science Foundation under grant CRSII2-147659 (FUPSOL II). C. Nehrbass-Ahles and F. Joos acknowledge support by the Swiss National Science Foundation. S.J. Phipps was supported under the Australian Research Council's Special Research Initiative for the Antarctic Gateway Partnership (Project ID SR140300001). C. Timmreck received funding from the German Federal Ministry of Education and Research (BMBF), research program "MiKliP" (FKZ: 01LP1517B) and the European Union FP7 project "STRATOCLIM" (FP7-ENV.2013.6.1-2; Project 603557). J. Jungclaus, A. Schurer and P. Braconnot received support from the Belmont/JPI-Climate Project PACMEDY (Paleo-Constraints on Monsoon Evolution and Dynamics; BMBF FKZ:01LP1607B, NERC: NE/P006752/1). J. Luterbacher acknowledge the German Science Foundation (DFG) project AFICHE (Attribution of forced and internal Chinese climate variability in the Common Era) and the Belmont/JPI-Climate Project INTEGRATE (An integrated data-model study of interactions between tropical monsoons and extra-tropical climate variability and extremes). K. Klein Goldewijk is supported by the Dutch NWO VENI grant no. 016.158.021and endorsed by the PAGES LandCover6k group. A. I. Shapiro acknowledges funding from the People Programme (Marie Curie Actions) of the European Union's Seventh Framework Programme (FP7/2007-2013) under REA grant agreement No. 624817. A Schurer was supported by the ERC funded project TITAN (EC-320691). J.F. González-Rouco acknowledges project ILModelS CGL2014-59644-R and R. Muscheler received support from the Swedish Research Council (grant DNR2013-8421). J. E. Smerdon was supported in part by US National Science Foundation grants AGS-1401400, AGS-1243204, and AGS-1602581 ( LDEO contribution #XXXX).

**Appendix A**

In this section we provide additional information on the derivation of the boundary conditions and recommendations for implementation in the individual models.

**A 1: Orbital parameters:**

Unless the orbital parameters are calculated based on the internal calendar, models should use the pre-calculated table that has been provided by (Schmidt et al., 2011) for the PMIP3 *past1000* simulations. The orbital parameters eccentricity, obliquity, and longitude of perihelion are calculated following Berger (1978).

**A2: Greenhouse gas forcing**

GHG ($CO_2$, $CH_4$, $N_2O$) concentrations are provided by Meinshausen et al. (2017) for the CMIP6 *historical* experiments. This data set has been extended to cover the entire CE (1 to 2014 CE). Data sets, and documentation are also available under: http://www.climatecollege.unimelb.edu.au/cmip6.

**A3: Volcanic forcing**

The eVolv2k ice core-inferred volcanic stratospheric sulfur injection from 500 BCE to 1900 CE (Toohey and Sigl, 2017) can be downloaded at http://cera-www.dkrz.de/WDCC/ui/EntryList.jsp?acronym=eVolv2k_v1.

The volcanic forcing package provided in the data supplement to this manuscript contains the eVolv2k data set and the EVA (Toohey et al., 2016) software available here: https://pmip4.lsce.ipsl.fr/doku.php/exp_design:lm. EVA contains a Fortran module and input data sets including the sulphate injection time-series, a Mie lookup table, and files specifying EVA parameter settings.

**A4: Solar forcing**

**A4.1: Derivation of magnetic flux and sunspot numbers from isotope data**

The [14]C-based scenarios are based on a recent reconstruction of the [14]C production rate by Roth and Joos (2013) from the INTCAL09 record (Reimer et al., 2009). First, it was converted to the heliospheric modulation potential, which parameterizes the energy spectrum of galactic cosmic rays (Usoskin et al., 2005). This was done with up-to-date models of radiocarbon production (Kovaltsov et al., 2012; Poluianov et al., 2016) and the geomagnetic field (Usoskin et al., 2016b). The open solar magnetic flux and the equivalent sunspot number were subsequently inferred from the modulation potential following the method of Krivova et al. (2007). Further details are described by Usoskin et al. (2014).

For [10]Be, the recent record, from the Antarctic Dome C site, by Baroni et al. (2015) has been used. This record features a correction for volcanic influence considering the [10]Be production model by Kovaltsov and Usoskin (2010) and the parameterization of the beryllium atmospheric transport by Heikkilä et al. (2009). The modulation potential was converted to sunspot numbers as for [14]C. The geomagnetic field was considered as IGRF (International Geomagnetic Reference Field; Thébault, et al., 2015) from 1900 CE and GEOMAG.9k (Usoskin et al., 2016b) before that. Because the snow accumulation rate is unknown, a constant accumulation rate was assumed when converting the concentrations to depositional flux. This

introduces a free scaling parameter, which is selected by equalizing the mean modulation potential between 880 and 1750 CE to that from the [14]C-based reconstruction of Usoskin et al. (2016b).

We use radionuclide data for the entire period before 1850 CE. This is done for two reasons. Firstly, this assures a single transition in the irradiance model configuration in 1850 CE, after when the CMIP6 historical record is to be used. Secondly, we avoid relying on sunspot observations, which are particularly uncertain before 1800 CE and presently the subject of intense debate (e.g., Clette et al. 2014, Usoskin et al. 2016a, Lockwood et al. 2016). The agreement with the exclusively cosmic ray-based reconstruction (Muscheler et al., 2016; Usoskin et al., 2014, 2016b) indicates that this approach does not introduce major discontinuities in the solar forcing record.

### A4.2: Transition from the pre-industrial SSI record to the recommended CMIP6 historical forcing

The reconstructions are combined with the solar forcing recommended for the CMIP6 historical experiments (Matthes et al., 2017). To achieve a smooth transition, both TSI and SSI are matched by rescaling them to the *historical* forcing near 1850 CE. The solar spectrum in 1855 CE in the CMIP6 solar forcing record (i.e. the first activity minimum covered by the CMIP6 record) is considered as a point of reference. The quiet Sun spectrum from the pre-1850 CE is scaled at each wavelength to fit this "reference spectrum". The procedure is illustrated in Figure 4 for the [14]C-based SATIRE reconstruction. The blue line shows the original reconstruction of the TSI and SSI in 3 broad spectral intervals (in the UV between 200 and 400 nm, in the visible at 400-700 nm and in the near-IR at 700-1200 nm wavelength). The cyan line is the same after rescaling to the CMIP6 *historical* quiet Sun spectrum (i.e. the recommended PMIP4 default). A consequence of the rescaling is that overall more radiation (about 0.3% of the total energy) comes at wavelengths below 700 nm compared to the original reconstruction, while the radiative flux above 700 nm is reduced by this amount.

### A4.3: Construction of atmospheric solar-related ozone variations for PMIP4

In order to simulate atmospheric (and in particular stratospheric) signals of solar irradiance variability, not only the irradiance variability itself but also ozone variations caused by it need to be accounted for. In PMIP4 simulations, models do not include interactive ozone and hence respective solar-induced variations in ozone need to be prescribed. As described in Section 4.4, we recommend to use average ozone data for preindustrial conditions, but to add an anomaly caused by the dependence of ozone on solar irradiance for any given time t:

$O_3(t) = O_3(t_{picontrol}) * (1 + 0.01 * A * \Delta J_{UV}(t))$.

$O_3(t_{picontrol})$ is the ozone mole fraction climatology available for the CMIP6 picontrol simulation ( the file "vmro3_input4MIPs_ozone_CMIP_UReading-CCMI-1-0_gr_185001-185012-clim.nc" can be found on input4MIPs (https://esgf-node.llnl.gov/search/input4mips/). *A* contains the linear regression coefficients of the ozone mole fraction with respect to changes in solar UV irradiance given in % ozone change per unit (i.e. 1 W/m$^2$) irradiance change in the spectral range from 200 to 320 nm (file available at https://pmip4.lsce.ipsl.fr/doku.php/data:solar). $\Delta J_{UV}(t)$ is the time dependent anomaly of UV irradiance (see section 4.4 of the main text) with respect to average preindustrial values. We recommend calculating the preindustrial average irradiance over the period 1.1.1850 to 28.1.1873 as recommended by Matthes et al. (2017).

The coefficients *A* have been calculated in the stratosphere using a multiple linear regression model (that includes a basis function for solar UV irradiance) fitted to the time series for CMIP6 historical ozone (taken from input4MIPs) over the period 1960-2011 (see Maycock et al. (2016a) for more details of method). While there are some uncertainties in the spatial structure and amplitude of the solar-ozone response diagnosed by multiple regression analysis, particularly at high latitudes in the stratosphere, we recommend this approach so as to be as consistent as possible with the representation of solar-induced ozone variations prescribed in models without interactive chemistry in the CMIP6 historical simulations.

Note, while $O_3(t_{picontrol})$ is given in three dimensions (latitude, longitude, pressure), $A$ is two-dimensional, i.e. it is provided as zonal average, and needs to be applied to all longitudes of the ozone climatology. It should also be noted that both the preindustrial ozone climatology and the correlation coefficients are given as an average annual cycle in terms of monthly means while the time resolution of the original irradiance data is as annual means for the period before 1850. In order to treat radiation and ozone consistently, the same interpolation of annual irradiance data to monthly mean data should be applied for the radiation calculations and the scaling of ozone fields. We recommend an amplitude-conserving interpolation as suggested by Sheng and Zwiers (1998).

**A4.4: Solar forcing data sets provided by PMIP4**

The forcing data sets are available from the PMIP4 web site: https://pmip4.lsce.ipsl.fr/doku.php/data:solar. We discriminate between the reconstructions derived using the SATIRE-M irradiance model (either [14]C or [10]Be –based) and the PMOD irradiance model ([14]C-based only). Note that the [14]C-based SATIRE-M data set scaled to the CMIP6 historical forcing is the recommended forcing for the PMIP4-CMIP6 *tier-1 past1000* experiment.

**A5: Land-Use Changes:**

These global gridded land-use forcing datasets are being developed as a contribution of the Land-Use Model Intercomparison Project (LUMIP) to link historical land-use data and future projections in a standard format required by climate models. This new generation of "land use harmonization" (LUH2) builds upon past work from CMIP5, and includes updated inputs, higher spatial resolution, more detailed land-use transitions, and the addition of important agricultural management layers. LUH2 has been extended in time to cover the pre-industrial millennium and the historical period (850 CE to 2015). Therefore PMIP4 *past1000* experiments use exactly the same data set as the CMIP6 *historical* experiment.

**A5.1: Derivation of the contribution by wood consumption**

The fraction of total wood demand that is used for durable goods is a function of GDP and varies from about 1% for subsistence-level GDP to about 15% of total demand at peak pre-fossil era GDPs (e.g. for the Netherlands around 1650 CE). For the period 850-1800 CE, total wood consumption is calculated as a function of baseline per-capita demand, a GDP-based consumption scalar, where higher GDP translates to higher per-capita consumption, and total country-level population from HYDE3.2 (Klein Goldewijk, 2016). For the baseline LUH2 scenarios, the national per capita wood harvest rates were multiplied by national scale factors that account for wood harvest processes. These scale factors are derived from the assumption that total global per capita rates of wood harvest increased by approximately a factor of two from current day rates to year 1800 rates based on estimates by Smil (2010). In the fossil energy era, which started in the late 18th century CE in some world regions, GDP and total energy consumption become uncoupled from wood demand. This uncoupling process varied greatly by country and over time. The final GDP-based wood consumption estimate is made at 1800 CE. Wood consumption is calculated for the period 1801-1920 CE using a linear interpolation of per capita wood harvest rates to the first historical estimates of global wood demand at 1920 CE (Zon and Sparhawk, 1923) and then computing the total national wood harvest demand by multiplying these per capita rates by the national population from HYDE3.2. The resulting wood consumption time series indicates strong declines in historical wood consumption over the 19th and early 20th centuries in most early-industrializing countries, whereas some countries continue to increase demand over the entire period (not shown). Within the LUH2 model, for the years 850-1850 CE, land cleared for agriculture is first used to satisfy wood harvest demands within each country before direct wood harvest occurs. From 1850-1920 CE, the fraction of land cleared for agriculture that is used towards meeting wood harvest demands is linearly decreased to 0 by 1920 CE. Additionally, for all years when wood harvest demands cannot be met for countries within Europe, the remaining wood harvest demand is spread across other European countries.

**A5.2: Land-use data set provided by PMIP4-CMIP6:**

The major attributes of the dataset include: Global domain with 0.25x0.25 degree resolution, annual land-use states, transitions, and gridded management layers, 12 land-use states including separation of primary and secondary natural vegetation into forest and non-forest sub-types, pasture into managed pasture and rangeland, and cropland into multiple crop functional types, over 100 different possible transitions per grid cell per year, including crop rotations; agriculture management layers including irrigation, fertilizer, and biofuel management. The CMIP6 Land Use Harmonization data set has been developed as part of the Land Use Model Intercomparison Project LUMIP (Lawrence at al., 2016) and can be downloaded from the LUMIP web site (http://luh.umd.edu/) and through Input4MIPs.

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

**Table 1:**

| Category | Experiment | Simulation years (single realisation) | Short name | extension |
|---|---|---|---|---|
| tier-1 | PMIP4-CMIP6 last millennium experiment using default forcings | 1000 (850 – 1849 CE) | past1000 | r\<K\>i1p1f1 |
| " | historical experiment using default CMIP6 forcings initialized from any past1000 simulation | 165 (1850 -2014 CE) | historical | r\<K\>i\<L\>p1f1 |
| tier-2 | PMIP4 last millennium experiment using alternative forcing data sets | 1000 (850 – 1849 CE) | past1000 | r\<K\>i1p1f\<N\> |
| tier-2 | PMIP4 last millennium experiment using single forcings | 1000 (850 – 1849 CE) | past1000-solaronly past1000-volconly | r\<K\>i1p1f\<N\> |
| tier-3 | PMIP4 last two millennia experiment | 1850 (1 – 1849 CE) | past2k | r\<K\>i1p1f\<N\> |
| " | CMIP6 historical experiment initialized from past2k | 165 (1850-2014 CE) | historical | r\<K\>i\<L\>p1f1 |
| " | PMIP4 volcanic cluster ensemble experiment (in cooperation with VolMIP) | 69 (1791-1849) | past1000-volc-cluster | r[1..3]i1p1f1 |
| " | PMIP4 last millennium experiment with interactive carbon cycle | 1000 | past1000esm | r\<K\>i1p1f1 |
| " | CMIP6 historical experiment with interactive carbon cycle initialized from esmPast1000 | 165 | esm-hist | r\<K\>i\<L\>p1f1 |

**Table 1:** List of experiments. In the right column the extension defines the ensemble member by the quad K, L, M, and N of integer indices for "realization" (r), "initialization" (i), "perturbed physics" (p), and "forcing (f). Modelling groups need to document the choices, in particular for initialization and forcing. Note that "f1" should be reserved for the default PMIP4-CMIP6 forcing.

**Table 2:**

| Feature | PMIP4 recommendation | Source |
|---|---|---|
| Orbital | Time-varying | Berger, 1978, Schmidt et al., 2011: https://wiki.lsce.ipsl.fr/pmip3/doku.php/ pmip3:design:lm:final#orbital_forcing |
| Greenhouse gases $CO_2$, $N_2O$, $CH_4$ | Time-varying, Same data set as historical | Meinshausen et al., 2017: http://www.climatecollege.unimelb.edu.au/cmip6 https://pcmdi.llnl.gov/search/input4mips/ |
| Volcanic forcing | Time-varying sulphur injections | Sigl et al., 2015; Toohey and Sigl, 2017: http://cera-www.dkrz.de/WDCC/ui/Compact.jsp?acronym=eVolv2k_v1 |
| Volcanic aerosol optical properties[1] | EVA module | Toohey et al., 2016: https://pmip4.lsce.ipsl.fr/doku.php/exp_design:lm |
| Solar irradiance | TSI and SSI time-varying | https://pmip4.lsce.ipsl.fr/doku.php/data:solar_satire |
| Ozone | Parameterization of solar-related variations | |
| Tropospheric aerosols | Methodology same as PiControl | |
| Vegetation | Methodology same as PiControl | |
| Land-cover changes | Same data set as historical | Lawrence et al., 2016; Hurtt et al., in prep. LUH2: http://luh.umd.edu/ https://pcmdi.llnl.gov/search/input4mips/ |

**Table 2:** Summary of boundary conditions for the PMIP4/CMIP6 "tier-1" *past1000* experiment.
[1]For models that need aerosol optical properties as forcing.

**Figure 1**

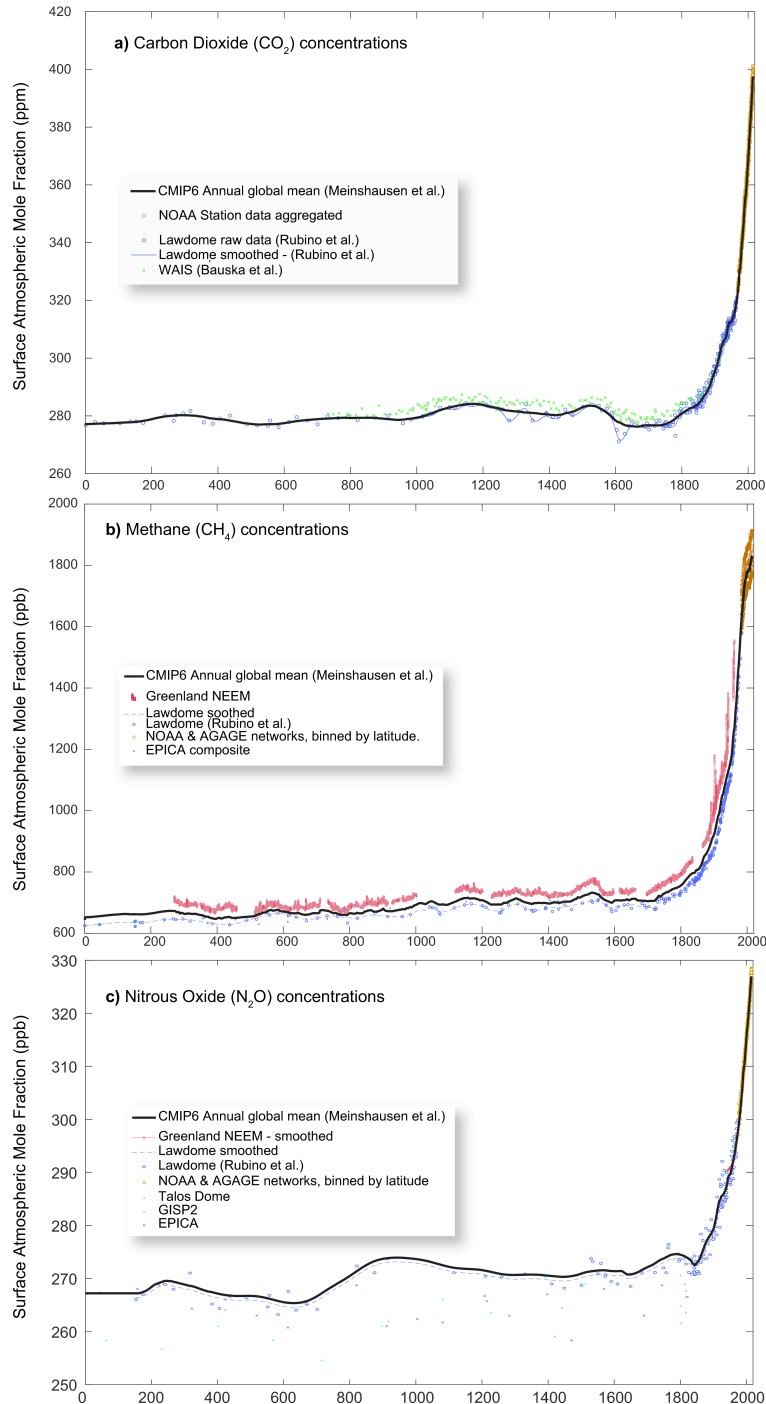

**Figure 1:** historical atmospheric surface concentrations from year 1BC to year 2014 CE of carbon dioxide, methane and nitrous oxide. The PMIP recommendation is to use GHG concentrations for *past1000* consistent with the *historical* CMIP6 runs. Here shown are global-mean concentrations of these fields (thick black line), in comparison with key Antarctic ice core and Greenland firn datasets (see legend). The latitudinal gradient for $CO_2$ is assumed zero before 1850 CE. For methane, NEEM and Law-Dome ice core data provides an indication of the latitudinal gradient during pre-industrial times, which is reflected in the extended CMIP6 dataset. $N_2O$ measurements from Antarctic ice cores vary substantially between studies. The extended CMIP6 dataset follows a smoothed version of the Law-Dome record.

**Figure 2**

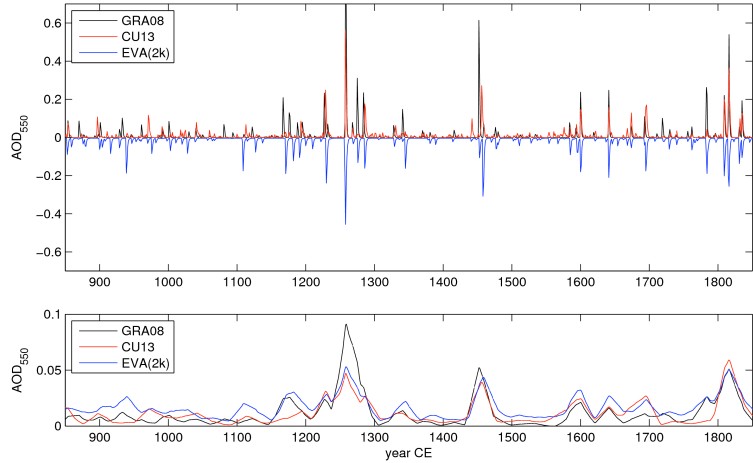

**Figure 2:** Reconstructions of volcanic forcing, 850-1850 CE, shown as global mean, mid-visible (550 nm) aerosol optical depth (AOD) as (top) annual means and (bottom) a smoothed time series after application of a 21-yr wide triangular filter (for visualisation). Reconstructions include the Gao et al., 2008 (GRA08), Crowley and Unterman 2013 (CU13) and the PMIP4 recommended forcing, EVA(2k). Note that the AOD in 1258 for the GRA08 reconstruction extends beyond the axis of the plot, with a value of approximately 1.05. AOD for the EVA(2k) reconstruction is shown on inverted axis in top panel for clarity.

**Figure 3**

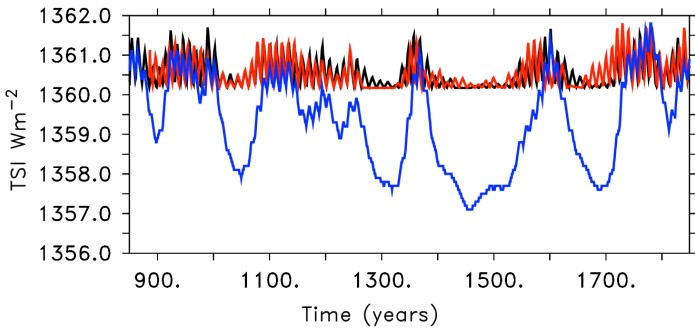

**Figure 3:** Reconstructions of Total Solar Irradiance based on two different isotope data sets and two different irradiance models. The [14]C- based reconstruction of sunspot numbers is converted to TSI using (black line) the SATIRE-M model, and (blue line) the updated Shapiro et al. (2011) model. The [10]Be-based TSI reconstruction is constructed using the SATIRE-M model (red line).

**Figure 4**

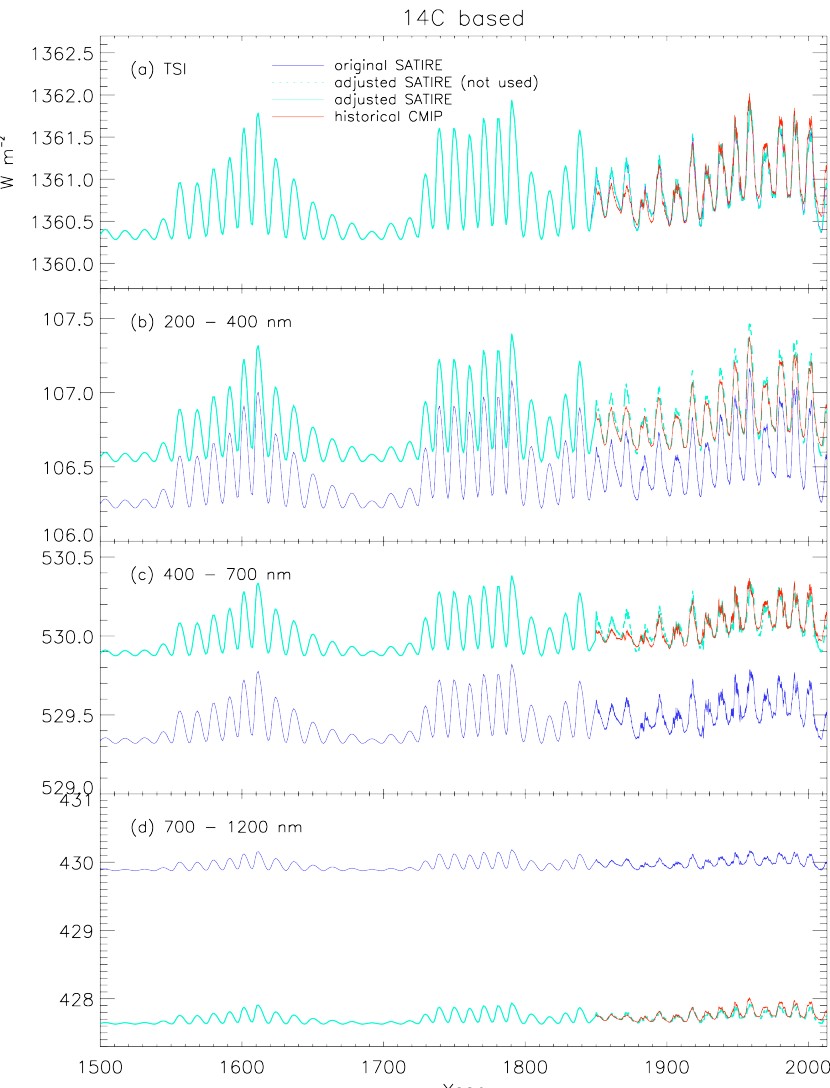

**Figure 4:** Adjustment of the [14]C/SATIRE-based reconstruction to the CMIP6 historical forcing (Matthes et al., 2017). TSI (a) and SSI (b - d) in 3 broad spectral intervals (in the UV between 200 and 400 nm, in the visible at 400-700 nm and in the near-IR at 700-1200 nm wavelength). The blue lines are the original [14]C/Satire based time series, the cyan lines represent the adjusted data, and the red line the CMIP6 forcing.

**Figure 5**

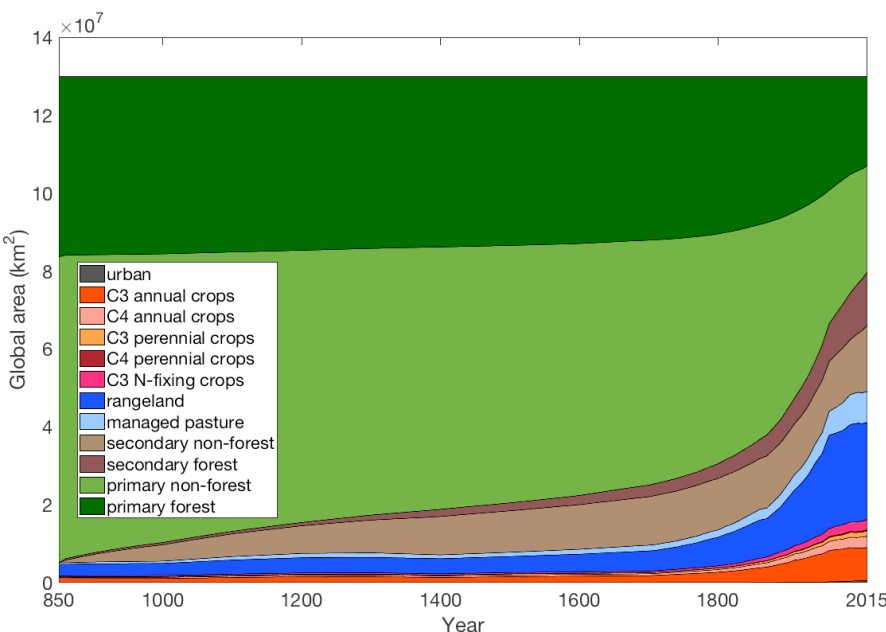

**Figure 5:** Evolution of various types of land-cover and land-use changes over the pre-industrial millennium.

