# Peer review of "The PMIP4 contribution to CMIP6 - Part 3: the Last Millennium, Scientific Objective and Experimental Design for the PMIP4 past1000 simulations"

_Geoscientific Model Development, 2016_

## Referee Comment (RC1) · Anonymous Referee #1 · 18 Jan 2017

Review Jungclaus et al. 'The PMIP4 contribution to CMIP6 - Part 3: the Last Millennium, Scientific Objective and Experimental Design for the PMIP4 past1000 simulations'

This paper provides a useful overview of the objective and setup of the planned PMIP4 simulations covering the preindustrial millennium. It is generally well written and clear, and I have only a few minor suggestions for improvement. This paper will serve the community well and I would recommend accepting it for publication in GMD.

Main comments

Introduction. In my view, the introduction could be improved by stating more clearly the objective of this paper and the added value compared to PMIP3-CMIP5. The main improvements relative to PMIP3 are summarized in Section 6 (Conclusions), but after reading the last two paragraphs of the introduction, it was not clear to me what the exact innovation is in the PMIP4 past1000 simulations compared to PMIP3. So I suggest to revise the main paragraph on page 4 (starting at line 5) to clarify this point.

Scope of the paper. The title and the introduction suggest that this paper is about the tier-1 past1000 experiments, but in fact also the forcings for the historical simulations are discussed. Is there a separate paper planned to explain the setup of the historical simulations in detail? If not, I would suggest to slightly modify the title to broaden the scope.

Section 3.3. I find the explanation of the tier-2 experiments rather vague. For instance, what is the time period to be covered in these simulations? Is it also the full 1000-year period of 850-1849 CE? Will the same initial conditions be used as in the tier-1 experiments? If the models are run in ensemble mode, what is the recommended number of ensemble members? I suggest making this section more specific.

Minor comments

Page 1, line 10. 'This is particularly acute for regional and sub-continental scales'. I suggest specifying what regions are of special interest here.

Page 1, line 15. 'preindustrial millennium'. Please explain here that you mean the 850-1849 CE period.

Page 6, line 11. 'a updated forcing datasets'. Remove 'a'.

Page 6, Section 3. Please briefly explain already here what the difference is between tier-2 and tier-3 experiments.

Page 6, line 26. I suggest mentioning here that the historical simulations cover 1850-2014 CE, and not in line 31.

Page 8, Section 3.4.2. Will the forcings for the past2K and past1000 experiments be identical for the period 850-1849 CE? I suggest clarifying this.

Page 8, Section 3.5. I suggest briefly explaining here or in the Table caption the meaning of the capitals N, M and L.

Page 9, Section 4.2. What is the difference in the GHG radiative forcing compared to PMIP3-CMIP5? Please explain.

Page 9, line, 8: 'Discrepancies in proxy-based temperature records'. Why are the temperature records mentioned here in the section on volcanic forcing? Please clarify.

Pages 10-11, Section 4.4. What is the difference in solar forcing compared to PMIP3? Please elaborate.

Page 11, line 26. Will ozone variations be provided by PMIP4 for the period 850-1849 CE? Please discuss.

Page 11, last line. Klein Goldewijk et al. 2016. The reference list only mentions Klein Goldewijk 2016, so without co-authors. Is the reference in the list incomplete?

Page 12. Section 4.5 discusses quite extensively the wood consumption. I wonder if this paper is the right place for this discussion, as it seems incompatible compared to level of detail in the rest of the manuscript. Wouldn't it fit better in a manuscript on LUMIP?

---

## Referee Comment (RC2) · Anonymous Referee #2 · 1 Feb 2017

In this manuscript the authors describe the major goals of the last millennium experiments within the forth phase of PMIP, and the experimental protocol that have been proposed to address them. This is an important well-organised initiative that will shed new light on both the internally driven and externally forced contributions to the climate of the last millennium, and will complement other additional efforts by the paleoclimate community (e.g. PAGES2K).

Therefore, I find the article timely and worthy of publication in Geoscientific Model Development. The paper is well written and the experimental protocol is well justified

and thoroughly explained. There are, however, some key choices of the experimental setup that could be better highlighted (see points below).

I thus recommend acceptance pending a few minor clarifications and comments that would need to be addressed.

Specific comments:

**1 I think that the article would benefit if the default forcings for the Tier1 experiments were more clearly synthetized, e.g. summarized in a Table and/or highlighted in the legends of the different figures. Otherwise, that key information is scattered throughout the text, and not always easy to find.**

**2 This article describes the third part of the PMIP4 contribution to CMIP6, but there is no mention to the other parts (are there more than three?), and how they complement with each other. A brief explanation in the introduction would be helpful.**

**3 I presume that the notation past1000 comes from the previous PMIP3 experimental protocol, and have been kept for coherence. I, however, think that the term is misleading, as it seems to suggest that the experiments cover the past millennium. But instead they target the "preindustrial" last millennium. I don't think that it's worth to change it now, but a more appropriate term could be considered in the future (e.g. preind1000).**

**4 [Page 3, lines 7-10] I would recommend rephrasing this sentence for clarity. For example, to something of the sort of ". . .the relative contribution of internal variability and external forcing factors to natural fluctuations in the Earth's climate system. . .".**

**5 [Page 4, line 15] Two other relevant articles that could be cited here are Lehner et al (2012) and Ortega et al ( 2015).**

**6 [Page 4, line 37] As it is written, it seems to imply that the MCA-LIA transition is only explained by these clusters of eruptions. But changes in solar irradiance most probably played some (minor) role. I suggest rephrasing to "Clusters of eruptions have been identified as the major contribution to the transition. . ."**

**7 [Page 6, lines 10-12] Remove "a" from "a updated". The final part could also be slightly rephrased to "a new generation of climate models in which the different forcings will be better represented". Also, it is not clear to me if this sentence refers exclusively to the changes in land-use, or to all the forcings previously described. If it's to all forcings, it might work better at the end of the paragraph (as the next sentence refers only to land-use changes).**

**8 [Page 5, line 13] Correct to "initiative".**

**9 [Page 7, lines 4-8] It is not totally clear to me from this paragraph whether there are two different sets of historical CMIP6 simulations according to their initial conditions (are they taken from picontrol experiments, past1000 experiments, or both?). Is that why you say that it will be possible to assess the impact of initial conditions on the climate of the 19th and 20th centuries?**

**10 [Page 13, line 21] Change to "impacted-related".**

**11 [Page 14, line 17] I suggest specifying "new climate reconstructions", to distinguish from forcing (reconstructions) just mentioned before.**

**12 [Page 14, first and second paragraph] These two collaborations with PAGES2K to investigate the past changes in the ocean circulation and hydroclimate are really important to bridge the existing gaps between models and paleo records. Will key variables for these model-data intercomparison studies, such as the AMOC and barotropic streamfunction and some drought severity indices, be consistently stored by the different modelling groups?**

---

## Editor Comment (EC1) · J. C. Hargreaves (Editor) · 15 Feb 2017

Please can you include details of where the model output data will be stored. Is it all to be uploaded to ESGF? If other databases are to be used, please give details and explain the terms of use. This basic information could be added to the data availability section. Secondly, please provide a list or table detailing the required output variables for the experiments. This could be added as an appendix or supplement.

---

## Author Response (AR2)

**Authors' response to Topical Editor Decision: Publish subject to minor revisions (Editor review) (25 Apr 2017) by Julia Hargreaves:**

5 For clarity, we reproduce the comments by the editor in blue/italic and provide answers in black. Changes to the manuscript are presented in bold face.

*Comments to the Author: Author Contribution paragraphs are often a good idea. In this case, where the authors have changed in the revision, I would like you to provide one. I think it may be positioned before the Acknowledgments.*

10 We have followed the editor's suggestion and included the following section before the Acknowledgements:

**Authors' contribution: J.H. Jungclaus, P. Braconnot, J. Cao, M. Evans, J.F. Gonzalez-Rouco, N. Krivova, J. Luterbacher, B. Otto-Bliesner, S.J. Phipps, G.A. Schmidt, J. Smerdon, M. Toohey, S. Wagner, and D. Zanchettin have designed the article, coordinated the writing and drafted the manuscript. The solar forcing reconstructions were**
15 **assembled and described by E. Bard, M. Baroni, T. Egorova, N. Krivova, R. Muscheler, E. Rozanov, H. Schmidt, W. Schmutz, A.I. Shapiro, S.K. Solanki, I.G. Usoskin, C.-J. Wu, K.L. Yeo. H. Schmidt and A.C. Maycock provided the section on solar-related ozone variations. M. Khodri, A.N. LeGrande, S.J. Lorenz, M. Sigl, C. Timmreck, and M. Toohey provided the volcanic forcing and the description of its implementation. L.P. Chini, G.C. Hurtt, J.O. Kaplan, K. Klein Goldewijk, and J. Pongratz contributed the land-use data set and its documentation. F. Joos, M.**
20 **Meinshausen, and C. Nehrbass-Ahles provided the section on greenhouse gas forcing. J. Cao, M. Evans, H. Goosse, J. Luterbacher, W. Man, A. Schurer, A. Moberg, Q. Zhang, and E. Zorita contributed to the section on specific analyses, the sections on the links between PMIP and other MIPs. All authors have contributed to the revisions and have approved of the final version of the manuscript.**

25 *Why is some of the data password protected? If you can provide a good reason for this, please also send me a password (or put it in the manuscript?!), so I can view the data.*

We apologize for not providing the password earlier. The password protection was installed only for the time that the review process is on-going because demands for changes in the forcing data sets could
30 have been brought-up. For this case, we wanted to be able to track down users in order to inform them about possible changes. We will lift the PW protection once the paper is accepted.
For the time being, the PW is "pmip4bc"

*For the model output, I like the way you have done it, specifying a minimum in the paper but also*
35 *including a file with the full set of desired variables. For people who can provide a bit more than the minimum, but not everything, is there any value in providing high frequency output for just part of the run? Would there be specific parts that would be of most interest?*

Thanks for the positive evaluation and the suggestion on specific time periods for high-frequency
40 output. We have discussed the matter in the group and came up with the following statement to be included in section:

**If storage of high-frequency output for the entire millennium should be too demanding, we recommend to concentrate efforts to three multi-decadal periods (in descending priority): 1. The**

**early 19th century (1790 to 1849 as focus period of VolMIP), and 2. the Maunder Minimum (1645 to 1715 CE) and 3.) the Medieval Climate Anomaly (1100 to 1170 CE) covering periods of high and low solar activity, respectively.**

*Note that the citation to Kageyama et al overview paper in the CMIP6 special issue is only to the GMDD paper in review. This is unlikely to improve since that paper is on hold pending the acceptance of this manuscript (and the other 2)!*

10   We will keep the citation to GMDD.

**Response to reviews of "The PMIP4 contribution to CMIP6 - Part 3: the Last Millennium, Scientific Objective and Experimental Design for the PMIP4 *past1000* simulations" by Jungclaus et al.**

For clarity, we reproduce the comments by the reviewers and the editor in blue/italic and provide answers in black. Changes to the manuscript are presented in bold face.

5  Reviewer 1:

*This paper provides a useful overview of the objective and setup of the planned PMIP4 simulations covering the preindustrial millennium. It is generally well written and clear, and I have only a few minor suggestions for improvement. This paper will serve the community well and I would recommend accepting it for publication in GMD.*

We thank the reviewer for his/her positive evaluation and suggestions that helped to improve the manuscript. In the
10  following, we address all the comments and suggestions.

*Introduction. In my view, the introduction could be improved by stating more clearly*
*the objective of this paper and the added value compared to PMIP3-CMIP5. The main improvements relative to PMIP3 are summarized in Section 6 (Conclusions), but after reading the last two paragraphs of the introduction, it was not clear to me what the exact innovation is in the PMIP4 past1000 simulations compared to PMIP3. So I suggest to revise the main*
15  *paragraph on page 4 (starting at line 5) to clarify this point.*

We have followed the reviewer's suggestion and included the following at the place suggested:

**Further progress is expected for CMIP6 and PMIP4. Models with higher spatial resolution will be available for long-term paleo simulations, which has the potential to improve the representation of mechanisms controlling regional variability and to alleviate biases in the mean state (e.g. Milinski et al., 2016). Newly added model components, for**
20  **example interactive chemistry and aerosol microphysics, will allow for more explicit representation of forcing-related processes in some models (LeGrande et al., 2016), and, as we outline below, improvements in forcing reconstructions regarding their accuracy and complexity will potentially lead to improved model data comparison. In addition, more stringent protocols for experimental set-ups and output data are implemented in the CMIP6 process, which also ensures a better interaction between related MIPs. For example, the PMIP4 past1000 experiment is closely related to**
25  **the more process-oriented suite of simulations in the Model Intercomparison Project on the climatic response to Volcanic forcing (VolMIP, Zanchettin et al., 2016).**

*Scope of the paper. The title and the introduction suggest that this paper is about the tier-1 past1000 experiments, but in fact also the forcings for the historical simulations are discussed. Is there a separate paper planned to explain the setup of the historical simulations in detail? If not, I would suggest to slightly modify the title to broaden the scope.*

30  The point is that it is very important to complement the past1000 simulations covering 850 to 1849 CE with historical simulations for the industrial period (1850 to 2014CE).

PMIP4 recommends strongly that these historical simulations will be carried out with the official CMIP6 historical forcing data sets that are documented in Eyring et al., GMD, 2016 and various other contributions to the GMD special issue. For some of the drivers, we are in the lucky situation that the forcings for the CMIP6 historicals have already been extended
35  back in time to cover either the entire CE (as for the GHG forcing), or the last millennium (850 to 2014 CE for the land-use forcing. For other forcings, e.g. solar we have made an effort to provide a smooth transition between the pre-industrial and the historical forcings. This is documented in the respective sections of our manuscript. Since the 1850-2014 simulations following the past1000s fulfil all the requirements for CMIP6 "historicals" we do not intend to describe and document them in this manuscript. We have however, underlined more clearly that it is mandatory to add a "historical" simulation that is

initialized with the 1849 conditions from past1000. Therefore, we have included the following statement at the end of the first paragraph of the introduction:

**We emphasize, that the *past1000* simulations must be complemented by *historical* simulations for 1850 to 2014 CE following the CMIP6 protocol and applying the CMIP6 external forcing for the industrial period (Eyring et al., 2016 and references therein).**

*Section 3.3. I find the explanation of the tier-2 experiments rather vague. For instance, what is the time period to be covered in these simulations? Is it also the full 1000- year period of 850-1849 CE? Will the same initial conditions be used as in the tier-1 experiments? If the models are run in ensemble mode, what is the recommended number of ensemble members? I suggest making this section more specific.*

We have added information on the simulation period etc. in the text

**The "tier-2" past1000 experiment should be set-up in a similar way as the "tier-1" past1000 simulation, i.e. the simulation should cover 850 to 1849 CE and the same initial conditions should be used.**

Regarding the ensemble size, we can probably not demand too much, therefore we have included the following sentence:

**While an ensemble size of ten has been shown to be desirable (Otto-Bliesner et al., 2016; Stevenson et al., 2016), we acknowledge that limits in computational resources or high computational demand of high-resolution models may prevent groups from conducting large ensembles.**

*Minor comments*

*Page 1, line 10. 'This is particularly acute for regional and sub-continental scales'. I suggest specifying what regions are of special interest here.*

At this point in the introduction we would prefer not to go into details regarding individual regions. We have, however, changed the sentence to better specify the issue of spatial inhomogeneity and included a reference to a recent paper by Gagen et al., who discuss sub-continental spatial variations over Europe:

**This is particularly acute for regional and sub-continental scales, where spatially heterogeneous variability modes potentially impact the climate signal (e.g., PAGES2k-PMIP3 Group, 2015; Luterbacher et al., 2016; Gagen et al., 2016).**

*Page 1, line 15. 'preindustrial millennium'. Please explain here that you mean the 850-1849 CE period.*

Done

*Page 6, line 11. 'a updated forcing datasets'. Remove 'a'.*

Done

*Page 6, Section 3. Please briefly explain already here what the difference is between tier-2 and tier-3 experiments.*

We included the following statement here:

In contrast to the PMIP3 protocol, PMIP4-CMIP6 recommends a single collection of external forcing data sets (the default forcing) in the "tier-1" experiments while encouraging exploration of forcing uncertainty as part of dedicated "tier-2" experiments. Whereas these "tier-2" experiments only differ in the characteristics and combination of the external drivers from the "tier-1" *past1000* experiment, additional "tier-3" experiments are designed to allow clusters of modelling groups to perform dedicated research by exploring either specific episodes or advancing the scope of the *past1000* experiments by extending them in time.

*Page 6, line 26. I suggest mentioning here that the historical simulations cover 1850- 2014 CE, and not in line 31.*

Done

*Page 8, Section 3.4.2. Will the forcings for the past2K and past1000 experiments be identical for the period 850-1849 CE? I suggest clarifying this.*

Yes, the forcing should be continuous. We have included the following statement after the first sentence of section 3.4.2. to clarify this:

15 **In fact, except for the land-use change forcing, all forcing reconstructions described above for the "tier-1" *past1000* experiment are available for the entire CE and the groups need to make sure that the same forcing is used for *past1000* and *past2k* during the period 850 to 1849 CE.**

*Page 8, Section 3.5. I suggest briefly explaining here or in the Table caption the meaning of the capitals N, M and L.*

20 We have expanded the description of the table in the text accordingly and changed slightly the nomenclature of the integers (now K, L, M, N):

**The experiments are defined by their short name (e.g., *past1000*) and an extension following the "ripf" classification, where "r" stands for "realization, "i" for initialization, "p" for perturbed physics, and "f" for forcing (Table 1). The letters r, i, p, and f are followed by integers K, L, M, and N, respectively. For example, different realisations within an**
25 **ensemble would have different values for "K" following the "r". To classify a simulation with a model with modified physical parameterization, one would vary the integer "M" after the "p". The experiments using the default forcing are defined by "f1", alternative or single forcing would be identified by a different integer value "N". CMIP6 *historical* simulations starting from a *past1000* run should vary the integer "L" after the "i". It is the responsibility of the modelling groups to document the choices and settings.**

*Page 9, Section 4.2. What is the difference in the GHG radiative forcing compared to PMIP3-CMIP5? Please explain.*

We have included the following addition:

**Differences between the new CMIP6 data set and previous estimates for CMIP5 are rather small (e.g., for global mean surface mixing rations see figure 9 in Meinshausen et al., 2016). The CMIP6 reconstruction offers better**
35 **representation of latitudinal and seasonal variations and we recommend using this data set for consistency throughout the CE.**

*Page 9, line, 8: 'Discrepancies in proxy-based temperature records'. Why are the temperature records mentioned here in the section on volcanic forcing? Please clarify.*

We did not make it clear enough that we mean the discrepancies in timing of cooling events in temperature reconstructions and the occurrence of the sulphate signal in the forcing reconstruction. We have rephrased this sentence:

**Discrepancies in the timing of volcanic events recorded in ice cores and short- term cooling events in proxy-based temperature records have been largely resolved by improvements in absolute dating of the ice core record (Sigl et al., 2015).**

*Pages 10-11, Section 4.4. What is the difference in solar forcing compared to PMIP3? Please elaborate.*

We have included a summary paragraph at the end of the section on solar forcing that describes what is new and more robust in PMIP4:

**In summary, PMIP4 provides three reconstructions of TSI and SSI from the most- up-to-date records of cosmogenic radioisotopes $^{14}$C and $^{10}$Be using a chain of models, all of which have been improved and updated since PMIP3. In contrast to CMIP3, for all provided reconstructions, total and spectral irradiance are computed in a self-consistent manner. In particular, the same model has been used to reconstruct irradiance from each radioisotope to allow an estimate of the uncertainty due to the effect of local conditions on their formation and deposition. Two irradiance reconstructions were obtained from $^{14}$C data using different irradiance models to allow for sensitivity experiments testing the response to the amplitude of the solar forcing. The default forcing for CMIP6-PMIP4 *past1000* is the $^{14}$C SATIRE-based reconstruction. The PMOD-based reconstruction provides an upper limit on the magnitude of the long-term changes in irradiance. Since the historical CMIP6 recommendation is an arithmetic average of two conceptually different models with significant differences in the SSI variability, special care has been taken to combine the PMIP4 data sets with the historical forcing. The approach we have chosen here allows for a smooth transition but might nevertheless produce some artefacts.**

*Page 11, line 26. Will ozone variations be provided by PMIP4 for the period 850-1849 CE? Please discuss.*

We have updated the section on solar-related ozone changes. For this part we received important input from Amanda Maycock, who we suggest to include as co-author. At the end of section 4.4, we now specify:

**Apart from the direct effect of changes in TSI and SSI, solar variability also affects stratospheric and mesospheric ozone abundances (e.g. Haigh, 1994) and can contribute significantly to the total stratospheric heating response. In climate models including interactive chemistry the photolysis scheme should adequately simulate the ozone response to variations in the UV part of SSI. CMIP6 models that do not include interactive chemistry should prescribe ozone variations consistent with the solar forcing and apply a scaling approach similar to the one recommended for the historical period (Matthes et al., 2016; Maycock et al., 2016b). It should be noted that solar-ozone regression coefficients as provided by Maycock (2016b) have been calculated with respect to the 10.7cm radio flux (F10.7), which is not available for the PMIP period. Hence we have re-performed the regression of the same ozone fields but with respect to solar UV irradiance averaged over the spectral range from 200 to 320 nm (see Appendix 4.3 for details). We recommend calculating time varying ozone input for PMIP4 by scaling these coefficients with the anomaly of the respective UV flux during the simulation period and add it to the CMIP6 preindustrial ozone climatology. The UV flux anomaly should accordingly be calculated with respect to the CMIP6 preindustrial irradiance data (Matthes et al., 2016).**

In addition, we included a new appendix 4.3 with a technical description of the ozone parameterization and its implementation.

*Page 11, last line. Klein Goldewijk et al. 2016. The reference list only mentions Klein Goldewijk 2016, so without co-authors. Is the reference in the list incomplete?*

We have corrected Klein Goldewijk et al., 2016 to Klein Goldewijk, 2016.

5 *Page 12. Section 4.5 discusses quite extensively the wood consumption. I wonder if this paper is the right place for this discussion, as it seems incompatible compared to level of detail in the rest of the manuscript. Wouldn't it fit better in a manuscript on LUMIP?*

We agree that the level of details on the reconstruction of wood consumption is a bit out of balance compared with the other ingredients to the forcing. We have shifted a larger part of this paragraph to the appendix section.

**Reviewer 2:**

*In this manuscript the authors describe the major goals of the last millennium experiments within the forth phase of PMIP, and the experimental protocol that have been proposed to address them. This is an important well-organised initiative that will shed new light on both the internally driven and externally forced contributions to the climate of the last millennium,*
15 *and will complement other additional efforts by the paleoclimate community (e.g. PAGES2K). Therefore, I find the article timely and worthy of publication in Geoscientific Model Development. The paper is well written and the experimental protocol is well justified and thoroughly explained. There are, however, some key choices of the experimental setup that could be better highlighted (see points below). I thus recommend*

*acceptance pending a few minor clarifications and comments that would need to be addressed.*

20 We thank the reviewer for his/her positive evaluation of our manuscript and the suggestions that we found helpful to improve the manuscript. In the following we will address all comments and suggestions.

*#1 I think that the article would benefit if the default forcings for the Tier1 experiments were more clearly synthetized, e.g. summarized in a Table and/or highlighted in the legends of the different figures. Otherwise, that key information is scattered*
25 *throughout the text, and not always easy to find.*

Thanks for pointing this out. We have followed the reviewer's suggestion and included such a table as table 2. The table includes also direct links to the respective forcing data repositories.

*#2 This article describes the third part of the PMIP4 contribution to CMIP6, but there is no mention to the other parts (are*
30 *there more than three?), and how they complement with each other. A brief explanation in the introduction would be helpful.*

We have clarified this point and provide a paragraph in the introduction that puts our paper into the context of the suite of PMIP4/CMIP6 papers.

**This paper is part of a suite of five manuscripts documenting the PMIP4 contributions to CMIP6. Kageyama et al. (2016a) provide an overview on the tier-1 experiments dedicated to CMIP. More specific information on the other**
35 **paleoclimate experiments as well as the design of additional tier-2 and tier-3 experiments are given in the contributions for the mid-Holocene and the previous interglacial by Otto-Bliesner et al. (2016), for the last glacial maximum by Kageyama et al. (2016b), for the mid-Pliocene warm period by Haywood et al. (2016), and the present manuscript on the last millennium. Our *past1000* manuscript is organized as follows....**

*#3 I presume that the notation past1000 comes from the previous PMIP3 experimental protocol, and have been kept for coherence. I, however, think that the term is misleading, as it seems to suggest that the experiments cover the past millennium. But instead they target the "preindustrial" last millennium. I don't think that it's worth to change it now, but a more appropriate term could be considered in the future (e.g. preind1000).*

We agree that the name past1000 could be misleading, but we are not in a position to change it now. Wes hall keep this issue for further discussion on transient simulations within PMIP.

*#4 [Page 3, lines 7-10] I would recommend rephrasing this sentence for clarity. For example, to something of the sort of ": : :the relative contribution of internal variability and external forcing factors to natural fluctuations in the Earth's climate system: : :". #5 [Page 4, line 15]*

done

*Two other relevant articles that could be cited here are Lehner et al (2012) and Ortega et al ( 2015).*

Included

*#6 [Page 4, line 37] As it is written, it seems to imply that the MCA-LIA transition is only explained by these clusters of eruptions. But changes in solar irradiance most probably played some (minor) role. I suggest rephrasing to "Clusters of eruptions have been identified as the major contribution to the transition: : :"*

done

*#7 [Page 6, lines 10-12] Remove "a" from "a updated". The final part could also be slightly rephrased to "a new generation of climate models in which the different forcings will be better represented". Also, it is not clear to me if this sentence refers exclusively to the changes in land-use, or to all the forcings previously described. If it's to all forcings, it might work better at the end of the paragraph (as the next sentence refers only to land-use changes).*

We agree that this did not fit very well. We have deleted the sentence on further progress in the land-use forcing (as it is irrelevant for the existing protocol) and moved the more general statement on the new generation models to the first paragraph of section 2:

**The PMIP4 experiments will revisit the questions regarding the relative role of external drivers using updated forcing datasets and a new generation of climate models, in which the different forcing will be better represented.**

*#8 [Page 5, line 13] Correct to "initiative".*

This sentence was deleted

*#9 [Page 7, lines 4-8] It is not totally clear to me from this paragraph whether there are two different sets of historical CMIP6 simulations according to their initial conditions (are they taken from picontrol experiments, past1000 experiments, or both?). Is that why you say that it will be possible to assess the impact of initial conditions on the climate of the 19th and 20th centuries?*

5   Indeed, there will be two sets of *historical* simulations. We designed the experimental set-up so that the 1850 to 2014 CE *historical* simulations should be identical in terms of forcing (and, as such serve as contributions to the CMIP6 multi-model ensemble of historical simulations), but differ in the initial conditions. On the other hand, we want to make sure that we have continuous simulations from the pre-industrial past to the historical period. We have slightly modified the text to make this point more clear:

10   **The standard PMIP4-CMIP6 *past1000* experiment applies the default forcing data set (see below) and is complemented by an *historical* (1850 – 2014 CE) simulation that uses the end state of the *past1000* simulation in 1850 CE for initialization and that follows the CMIP6 protocol (Eyring et al., 2016). This procedure provides a consistent data set for past and present climate variations. Comparing historical simulations initialized from a *piControl* run (the CMIP6 default) with those starting from 1849 CE conditions from *past1000* serves to assess the impact of initial**

15   **conditions on the evolution of the 19$^{th}$ and 20$^{th}$ century climate.**

*#10 [Page 13, line 21] Change to "impacted-related".*

Done

20   *#11 [Page 14, line 17] I suggest specifying "new climate reconstructions", to distinguish from forcing (reconstructions) just mentioned before.*

We changed this to:

**...new reconstructions of climate variables...**

to emphasize that it is not only about temperature.

*#12 [Page 14, first and second paragraph] These two collaborations with PAGES2K to investigate the past changes in the ocean circulation and hydroclimate are really important to bridge the existing gaps between models and paleo records. Will key variables for these model-data intercomparison studies, such as the AMOC and barotropic streamfunction and some drought severity indices, be consistently stored by the different modelling groups?*

30   Yes, this is a major effort that has been done in the CMIP6 community. We have added the following statement to the end of section 5.

**PMIP has provided to CMIP6 a comprehensive list of output variables that includes all necessary variables for analyses of atmospheric, oceanic and land- surface processes (see section 3.6). CMIP6 will make sure that all groups store the output variables in a consistent way (see https://earthsystemcog.org/projects/wip/CMIP6DataRequest).**

Please see also our response to the editor's comments below.

**Comments by the Editor (J.C. Hargreaves):**

*Please can you include details of where the model output data will be stored. Is it all to be uploaded to ESGF? If other databases are to be used, please give details and explain the terms of use. This basic information could be added to the data availability section. Secondly, please provide a list or table detailing the required output variables for the experiments. This*

5  *could be added as an appendix or supplement.*

We have expanded the section on variable selection and data distribution and moved it to a more central place in the main text (section 3.6 of the new version). We include more information on data distribution and the list of variables requested by PMIP4. However, we are facing the problem that we are not able to provide a definite and final list of variables including approval by the CMIP6 Infrastructure panel. The request PMIP has submitted to CMIP6 (**http://clipc-**

10  **services.ceda.ac.uk/dreq/u/PMIP.html**) is processes by WIP into official CMIP6 data request tables. The most up-to-date version can be downloaded here:

**proj.badc.rl.ac.uk/svn/exarch/CMIP6dreq/tags/latest/dreqPy/docs/CMIP6_MIP_tables.xlsx**

We follow the editor's request to include this list as supplement, but we must point out that this list is "work in progress" with the most recent update just two days ago. We expect that the list contains inconsistencies that shall be removed in

15  upcoming versions.

We have modified the manuscript and included a new section 3.6:

**3.6 Output variables and data distribution**
**The "tier-1"** *past1000* **simulation is part of the CMIP6 experiment family and output data will be distributed through**

20  **the official CMIP6 channels via the Earth System Grid Federation (ESGF,**
**https://earthsystemcog.org/projects/wip/CMIP6DataRequest).**
**Data from PMIP4-only "tier-2" and "tier-3" simulations must be processed following the same standards for data**
**processing (e.g. CMOR standards) and should be distributed via ESGF. Modelling groups producing these**
**simulations are responsible to secure suitable space on ESGF nodes.**

25  **Groups contributing** *past1000* **simulations to CMIP6-PMIP4 should ideally deliver the entire set defined in the data**
**request. However, an important issue for long-term simulations such as** *past1000* **is storage demand for high-**
**frequency output. As a minimum, we ask for a subset of two-dimensional daily variables that allow investigations on**
**extreme events and particular dynamical features, including near surface air temperature (tas), daily maximum near**
**surface air temperature (tasmax), daily minimum near surface air temperature (tasmin), daily maximum near-**

30  **surface wind speed (sfcWindmax), precipitation (pr), 500 hPa geopotential (zg500), daily maximum hourly**
**precipitation Rate (prhmax).**
**Groups participating in PMIP and VolMIP should pay attention to the new diagnostics of volcanic instantaneous**
**radiative forcing defined by VolMIP, whose calculation is recommended for some major volcanic events simulated in**
**the** *past1000* **experiment (for details, see Zanchettin et al., 2016). Groups that run the PMIP4-CMIP6 experiments**

35  **with the carbon cycle enabled should pay attention to the output variables requested by OCMIP and C4MIP.**
**The list of variables requested by PMIP for the PMIP4-CMIP6 palaeoclimate experiments can be found here:**
**http://clipc-services.ceda.ac.uk/dreq/u/PMIP.html.**
**This request is presently processed by the CMIP6 Working Group for Coupled Modeling Infrastructure Panel (WIP)**
**into tables, which define the variables included in the data request to the modelling groups for data to be contributed**

40  **to the archive. The most up-to-date list including all variables requested for CMIP6 can be found at the WIP site:**
**proj.badc.rl.ac.uk/svn/exarch/CMIP6dreq/tags/latest/dreqPy/docs/CMIP6_MIP_tables.xlsx**
**The last two columns in each row list MIPs associated with each variable. The first column in this pair lists the MIPs**
**which are requesting the variable in one or more experiments. The second column lists the MIPs proposing**
**experiments in which this variable is requested.**

45  **As supplementary to this manuscript we provide version 1.00.05 (April 2017) of the table. We note, however, that this**
**document is still in development and inconsistencies may still exist.**

In addition, we added the following sentence to the "Data Availability" section:

**The results of the experiments described here will be distributed via the Earth System Grid Federation (ESGF, https://earthsystemcog.org/projects/wip/CMIP6DataRequest), as described together with the requested output variables in section 3.6.**

**Further unsolicited changes**

We have refined the description of the solar forcing at several places and have collaborated with Prof. Raimund Muscheler (Lund University). We received important input from R. Muscheler regarding the methodology presented in section 4.4. In particular, the comparison between his recently published neutron-monitor based estimate for the solar modulation has
10 increased confidence in our solar forcing reconstruction.
We would like to include R. Muscheler as co-author in the revised version of our manuscript

We have further elaborated on the parameterization of solar-related ozone changes. We received important input from Dr. Amanda Maycock (University of Leeds), who has also been instrumental in defining the respective forcing for the CMIP6 historical experiments. We therefore wish to include A. Maycock as co-author in the revised version of our manuscript.

[revised manuscript text omitted]

Johann Jungclaus 11.4.2017 16:52

Johann Jungclaus 11.4.2017 16:52

Johann Jungclaus 11.4.2017 16:52

Johann Jungclaus 11.4.2017 16:52

Johann Jungclaus 17.3.2017 13:46

Johann Jungclaus 11.4.2017 16:52

Johann Jungclaus 17.3.2017 13:46

Johann Jungclaus 11.4.2017 16:52

Johann Jungclaus 17.3.2017 13:46

Johann Jungclaus 17.3.2017 13:46

Johann Jungclaus 17.3.2017 13:46

Johann Jungclaus 11.4.2017 16:53

Johann Jungclaus 17.3.2017 10:55

Johann Jungclaus 17.3.2017 13:46

Johann Jungclaus 17.3.2017 13:46

Johann Jungclaus 17.3.2017 13:46

Johann Jungclaus 11.4.2017 16:52

Johann Jungclaus 17.3.2017 13:46

Johann Jungclaus 17.3.2017 13:46

Johann Jungclaus 17.3.2017 13:46

Johann Jungclaus 11.4.2017 16:52

Johann Jungclaus 11.4.2017 16:53

**Table 2:**

| Feature | PMIP4 recommendation | Source |
|---|---|---|
| Orbital | Time-varying | Berger, 1978, Schmidt et al., 2011: https://wiki.lsce.ipsl.fr/pmip3/doku.php/pmip3:design:lm:final#orbital_forcing |
| Greenhouse gases $CO_2$, $N_2O$, $CH_4$ | Time-varying, Same data set as historical | Meinshausen et al., 2016: http://www.climatecollege.unimelb.edu.au/cmip6 https://pcmdi.llnl.gov/search/input4mips/ |
| Volcanic forcing | Time-varying sulphur injections | Sigl et al., 2015; Toohey and Sigl, 2016: http://cera-www.dkrz.de/WDCC/ui/Compact.jsp?acronym=eVolv2k_v1 |
| Volcanic aerosol optical properties[1] | EVA module | Toohey et al., 2016: https://pmip4.lsce.ipsl.fr/doku.php/exp_design:lm |
| Solar irradiance | TSI and SSI time-varying | https://pmip4.lsce.ipsl.fr/doku.php/data:solar_satire |
| Ozone | Parameterization of solar-related variations | |
| Tropospheric aerosols | Methodology same as PiControl | |
| Vegetation | Methodology same as PiControl | |
| Land-cover changes | Same data set as historical | Lawrence et al., 2016; Hurtt et al., in prep. LUH2: http://luh.umd.edu/ |

**Table 2:** Summary of boundary conditions for the PMIP4/CMIP6 "tier-1" *past1000* experiment.
[1]For models that need aerosol optical properties as forcing.

---

## Author Response (AR3)

Response to GMD editors' comments on "The PMIP4 contribution to CMIP6 - Part 3: the Last Millennium, Scientific Objective and Experimental Design for the PMIP4 *past1000* simulations" by Jungclaus et al.

For clarity, we reproduce the comments by the reviewers and the editor in blue/italic and provide answers in black. Changes to the manuscript are presented in bold face.

**Comments to the authors by J. Hargraves:**

I have managed to access the forcing data through the password protected site! Thus, the only addition I have to the issues common to all the PMIP experiment description papers is that you should email me to let me know the password has been lifted so that I can check this before I finally accept the paper.

The password protection has been lifted and the data can be downloaded without restrictions (see, for example: https://pmip4.lsce.ipsl.fr/doku.php/data:solar\_satire)

Aside from that, there are just the queries raised by me, James Annan and Didier Roche, editors of the 3 PMIP papers, which I already emailed to you, Masa and Bette. For convenience and transparency, these queries are repeated here below.

Sorry for the long time that the I-G and LM PMIP papers have been with the editors. We want to make sure we are being consistent across the papers. We have all looked through the manuscripts and found a number of things that need clarification, or that are inconsistent across the papers. We're emailing you all today so that you can start to coordinate your response, and then you should expect that most of these comments will also be included in the individual editor's revision for each manuscript.

Thanks for the coordinated view on the manuscripts. Your comments are very helpful for certain clarifications across the PMIP papers. We have coordinated the responses among the principal authors.

Some papers mention the entry cards for CMIP, but others do not. Some bring in the "Tier" nomenclature without defining it, while the LGM discusses "sensitivity" experiments, leaving it uncertain as to how these fit into the CMIP system.

\* All papers should explain the entry card system and contain full and consistent information on the documentation requirement for the runs. Many groups will do all the Tier 1 runs, so it really will be very confusing if the documentation requirements are different! All the papers should make sure to define CMIP concepts such as "Tier 1".

We have now categorized all experiments in the PMIP4 papers as Tier 1, Tier 2 and Tier 3. The concept of Tier is introduced in the CMIP6 overview paper (Eyring et al., 2016), with Tier 1 having the highest priority. Within PMIP4, the Tier 1 experiments are those, which are Tier 1 for CMIP6 as well. They are also reference experiments for Tier 2 and Tier 3 experiments described for each period in the PMIP4 GMD manuscripts, which will be made clearer in the revised versions (especially for the LGM sensitivity experiments, which will be categorized as Tier 2). In the PMIP4-CMIP6 overview paper, we introduce the concept of PMIP4 entry card, capitalizing on PMIP's previous experiments. These (the *midHolocene* and *lgm* experiments) are special PMIP4-CMIP6 Tier 1 experiments. We consider that at least one of these experiments must be performed by the modelling groups to be part of PMIP4-CMIP6, because they will allow us to monitor the progress from the previous phases of PMIP and CMIP. This is explained in the

overview paper, but will be recalled in the PMIP4 papers. We want to make it clear that the Tier2 and 3 experiments absolutely require the corresponding Tier 1 experiment for their analysis, so the groups must perform the Tier-1 experiment first.

In the past1000 manuscript we have modified the text:

This paper is part of a suite of five manuscripts documenting the PMIP contributions to CMIP6. Kageyama et al. (2016) provide an overview on the five selected time periods and the experiments. More specific information are given in the contributions for the mid-Holocene (*midHolocene*) and the previous interglacial (*lig127k*) by Otto-Bliesner et al. (2016), for the last glacial maximum (*lgm*) by Kageyama et al. (2017), and for the mid-Pliocene warm period (*midPliocene*) by Haywood et al. (2016), and the present manuscript on the last millennium (*past1000*). PMIP has adopted the CMIP6 categorization where the highest-priority experiments are classified as "tier-1", whereas additional sensitivity experiments or dedicated studies are "tier-2" or "tier-3". The standard experiments for the five periods are all ranked "tier-1". Modelling groups are not obliged to run all PMIP4-CMIP6 experiments. It is mandatory, however, for all participating groups to run at least one of the experiments that were run in previous phases of PMIP (i.e., *midHolocene* or *lgm*).

In the past1000 manuscript we classified all experiments (sections 3.2 to 3.4) as "tier-1", "tier-2", or "tier-3". This is also reflected in the updated table 1:

| Category Experiment                                                                                                              |                                                                                   | Simulation years
(single realisation) | Short name                              | extension             |
|----------------------------------------------------------------------------------------------------------------------------------|-----------------------------------------------------------------------------------|------------------------------------------|-----------------------------------------|-----------------------|
| tier-1                                                                                                                           | PMIP4-CMIP6 last millennium
experiment using default
forcings               | 1000
(850 - 1849 CE)                  | past1000                                | r <k>i1p1f1</k>       |
|  <li>historical experiment using
default CMIP6 forcings</li> <li>initialized from any past1000
simulation</li>  |                                                                                   | 165
(1850 -2014 CE)                   | historical                              | r <k>i<l>p1f1</l></k> |
| tier-2                                                                                                                           | PMIP4 last millennium
tier-2 experiment using alternative
forcing data sets |                                          | past1000                                | r <k>i1p1f<n></n></k> |
| tier-2 PMIP4 last millennium experiment using single forcings                                                                    |                                                                                   | 1000
(850 - 1849 CE)                  | past1000-solaronly
past1000-volconly | r <k>i1p1f<n></n></k> |
| tier-3                                                                                                                           | PMIP4 last two millennia
experiment                                            | 1850
(1 – 1849 CE)                    | past2k                                  | r <k>i1p1f<n></n></k> |
| "                                                                                                                                | CMIP6 historical experiment
initialized from past2k                            | 165
(1850-2014 CE)                    | historical                              | r <k>i<l>p1f1</l></k> |
| " PMIP4 volcanic cluster ensemble
experiment (in cooperation with
VolMIP)                                                  |                                                                                   | 69
(1791-1849)                        | past1000-volc-cluster                   | r[13]i1p1f1           |
| " PMIP4 last millennium
" experiment with interactive
carbon cycle                                                         |                                                                                   | 1000                                     | past1000esm                             | r <k>i1p1f1</k>       |
| CMIP6 historical experimer
" with interactive carbon cycl
initialized from esmPast100                                      |                                                                                   | 165                                      | esm-hist                                | r <k>i<l>p1f1</l></k> |

The documentation requirement seems to be different in all the papers (and the LGM is actually inconsistent with the overview paper). The documentation requirement is really presented as part of the protocol, so should be included in each of the individual papers.

The overview paper will insist on the importance of documenting the simulations. The specificities for each simulation are detailed in each paper, since they depend on the forcings for each experiment. This includes the documentation on spin-up and equilibrium as detailed below.

We have streamlined the documentation requirements and each manuscript now contains a section "Documentation". The specifications are, of course, different for the different time periods and experiments: For the past1000 paper, information on model drift should be given for the spin-up run that has been used to arrive at the initial conditions for the start of the actual transient experiment (see answer below).

**3.6 Documenting the simulations**

The modelling groups are responsible for a comprehensive documentation of the model system and the experiments. A PMIP4 special issue in GMD and Climate of the Past has been opened where the groups are encouraged to publish these documentations. The documentation should include:

- The model version and specifications, like interactive vegetation or interactive aerosol modules etc.
- A link to the DECK experiments performed with this model version
- Specification of the forcing data sets used and their implementation in the model
- A documentation of the spin-up strategy to arrive at 850 CE (1 CE for past2k) initial conditions. We request providing information on drift in key variables for a few hundred years at the end of the spin-up and the beginning of the actual experiment. These variables are:
  - globally and annually averaged SSTs
  - deep ocean temperatures (global and annual average over depths below 2500m)
  - deep ocean salinity (global and annual average over depths below 2500m)
  - top of atmosphere energy budget (global and annual average)
  - surface energy budget (global and annual average)
  - northern sea-ice (annual average over northern hemisphere)
  - southern sea-ice (annual average over southern hemisphere)
  - northern surface air temperature (annual average over northern hemisphere)
  - southern surface air temperature (annual average over southern hemisphere)
  - Atlantic Meridional Overturning Circulation (maximum overturning in the North Atlantic basin below 500 m)
  - carbon budget by the biosphere.

We are all confused about what the comment that groups are responsible for finding their own ESGF space for Tier2-3 experiments means in practice, and note that there is no indication as to whether the LGM sensitivity experiments should be uploaded or not.

\* Please clarify what, in practice, this group responsibility means. Will modellers be able to upload Tier 2 and 3 experiments to ESGF or not, and how are the LGM sensitivity runs to be made available?

On the PMIP side, we are taking all necessary measures so that the PMIP Tier-2 and Tier-3 output can be uploaded and published on the ESGF distributed network. However, LSCE, who is

coordinating the database, cannot provide the disk space for all modelling groups as it did until PMIP2, and for some of the groups in PMIP3. This is what we meant by the above statements. Modelling groups participating to PMIP will have to coordinate with their national ESGF node to upload their data on ESGF, as they will do for their PMIP4-CMIP6 Tier-1 data. The LGM sensitivity runs are now included in the list of Tier-2 experiments.

Are all boundary conditions to be uploaded to ESGF? This is promised in the LGM paper, but not in the others.

\* Please make it clear whether all boundary conditions will be uploaded to ESGF. It would be best if they were, as the current situation where everything is available on the PMIP website is suboptimal (it seems to me OK for modellers doing the runs now, but it is not future-proof: web addresses change!).

We have put a link in the ESGF/Input4MIPs reference document providing the connection to the PMIP4 web site. This way it is also done for other CMIP6 experiments (see the reference document entries for FAFMIP, OMIP, VolMIP etc.). However, owing to our priority to make the data sets available as fast as possible, we didn't have resources to make the data sets fully Input4MIPs compliant.

For the "past1000" experiment the forcing data sets for historical land-use/land cover change and greenhouse gases are identical with those for the CMIP6 "historicals" (i.e. just extensions in time into the Common Era) and are already available in Input4MIPs and accessible via ESGF. We have included this information in the text:

**All forcing data sets and the EVA tool for producing aerosol optical properties can be accessed via the PMIP4 *past1000* web page:**

https://pmip4.lsce.ipsl.fr/doku.php/exp\_design:lm. The forcing data sets provided exclusively for the *past1000* simulations (orbital, solar, volcanic), can be downloaded directly from the PMIP4 repository. They will be made available also through Input4MIPs (https://esgf-node.llnl.gov/projects/input4mips/, see the living document "Input4MIPs summary"). The CMIP6 historical forcing data sets that provide extensions into the Common Era (GHG, land-use) and that are documented in individual contributions to the CMIP6 GMD special issue are already accessible through ESGF (https://esgfnode.llnl.gov/search/input4mips/) or via links to the originators' web pages (see "Input4MIPs summary"). The results of the experiments described here will be distributed via the Earth System Grid Federation (ESGF,

https://earthsystemcog.org/projects/wip/CMIP6DataRequest), as described together with the requested output variables in section 3.7.

The LGM experiment requires 100 years of equilibrated run to be stored on ESGF, and defines which variables are to be used by participating groups to ensure sufficient equilibrium is attained before the other experiments are started. There is less detail on this in the other papers.

\* Please can all experiments require the 100 years of equilibrated run on ESGF. If possible, please define a common metric for assessing equilibrium for all the runs. If there are good reasons why different metrics should be used for the different experiments, please provide an explanation and guidance for the modellers.

For the past1000 experiments the issue is to run the spin-up long enough to provide proper

initial conditions for the following transient simulation. We have specified this in sections 3.1 ("Initial state") and 3.6 "Documentation" (see comments above) follows:

To provide initial conditions for the simulations, it is recommended that a spin-up simulation is performed departing from the CMIP6 *piControl* experiment with all forcing parameters set to ~850 CE values. The length of this spin-up simulation will be model- and resource- dependent. However, it should be long enough to minimize at least surface climate trends (Gregory, 2010). The spin-up has to be documented and this should include information on a few key variables (see section 3.6). The spin-up should be consistent with the *piControl* (for example, regarding a background volcanic aerosol level), and should include appropriate anthropogenic modifications to land use/land cover characteristics (as for the *piControl* simulation; see Eyring et al., 2016).

**The PMIP4 contribution to CMIP6 - Part 3: the Last Millennium, Scientific Objective and Experimental Design for the PMIP4 past1000 simulations**

- Johann H. Jungclaus1, Edouard Bard2, Mélanie Baroni2, Pascale Braconnot3, Jian Cao4, Louise P. Chini5, Tania Egorova6,7, Michael Evans8, J. Fidel González-Rouco9, Hugues Goosse10, George C. Hurtt5, Fortunat Joos11, Jed O. Kaplan12, Myriam Khodri13, Kees Klein Goldewijk14,15, Natalie Krivova16, Allegra N. LeGrande17, Stephan J. Lorenz1, Jűrg Luterbacher18,19, Wenmin Man20, Amanda C. Maycock21, Malte Meinshausen22,23, Anders Moberg24, Raimund Muscheler25, Christoph Nehrbass-Ahles11, Bette I. Otto-Bliesner26, Steven J. Phipps27, Julia Pongratz1, Eugene Rozanov6,7, Gavin A. Schmidt17, Hauke Schmidt1, Werner Schmutz6, Andrew Schurer28, Alexander I. Shapiro16, Michael Sigl29,30, Jason E. Smerdon31, Sami K. Solanki16, Claudia Timmreck1, Matthew Toohey32, Ilya G. Usoskin33, Sebastian Wagner34, Chi-Ju Wu16, Kok Leng Yeo16, Davide Zanchettin35, Qiong Zhang24, and Eduardo Zorita34 5
- and Eduardo Zorita34

15

- 1Max Planck Institut für Meteorologie, Hamburg, Germany

[revised manuscript text omitted]

| Johann Jungclaus 29.6.2017 17:53                                                                                         |
|--------------------------------------------------------------------------------------------------------------------------|
| Johann Jungclaus 29.6.2017 17:47                                                                                         |
| Johann Jungclaus 29.6.2017 17:47                                                                                         |
| Johann Jungclaus 29.6.2017 17:54                                                                                         |
| Johann Jungclaus 29.6.2017 17:48                                                                                         |
as well as the design of additional tier-2 and
tier-3 experiments |
| Johann Jungclaus 29.6.2017 17:52                                                                                         |
| Johann Jungclaus 29.6.2017 17:52                                                                                         |
| Johann Jungclaus 29.6.2017 18:01                                                                                         |
| Johann Jungclaus 29.6.2017 17:52                                                                                         |
| Johann Jungclaus 29.6.2017 17:53                                                                                         |
| Johann Jungclaus 29.6.2017 18:01                                                                                         |
| Johann Jungclaus 29.6.2017 18:01                                                                                         |
|                                                                                                                          |

Our *past1000* manuscript is organized as follows. In section 2, we review the major forcing agents for climate evolution during the CE in the light of previous simulations of the past. Section 3 describes the experimental protocols for the tier-1 to tier-3 categorized experiments. Section 4 describes the derivations and the characteristics of the forcing boundary conditions. Section 5 discusses the relations between the PMIP experiments and the overarching research questions of CMIP6 and links

5 to other MIPs. Section 6 provides a concluding discussion.

**2 Drivers of climate variations during the CE**

The major forcing agents during the pre-industrial millennium are changes in orbital parameters, solar irradiance, stratospheric aerosols of volcanic origin, and greenhouse gas (GHG) concentrations. Additional anthropogenic impacts arise from aerosol emissions and changes in land-surface properties as a result of land use (e.g. Pongratz et al., 2009; Kaplan et al., 2011). External drivers affect the climate system in several ways, ranging from millennial-scale trends, such as those induced by changing orbital parameters, to the response of relatively short-lived disturbances of the radiative balance, as in the case of volcanic activity. Additionally, feedbacks internal to the climate system may amplify, delay, or prolong the effect of forcing (e.g., Shindell et al., 2001; Swingedouw et al., 2011; Zanchettin et al., 2012). The PMIP4 experiments will revisit

[revised manuscript text omitted]
                                    | Time-varying,        | Meinshausen et al., 201 7 ;                    |
|   | CO 2 , N 2 O, CH 4 | Same data set as     | http://www.climatecollege.unimelb.edu.au/cmip6        |
|   |                                                     | historical           | https://pcmdi.llnl.gov/search/input4mips/             |
|   | Volcanic forcing                                    | Time-varying sulphur | Sigl et al., 2015; Toohey and Sigl, 2017;             |
|   |                                                     | injections           | http://cera-                                          |
|   |                                                     |                      | www.dkrz.de/WDCC/ui/Compact.jsp?acronym=eVolv2k_v1    |
|   | Volcanic aerosol                                    | EVA module           | Toohey et al., 2016:                                  |
|   | optical properties 1                     |                      | https://pmip4.lsce.ipsl.fr/doku.php/exp_design:lm     |
|   | Solar irradiance                                    | TSI and SSI time-    | https://pmip4.lsce.ipsl.fr/doku.php/data:solar_satire |
|   |                                                     | varying              |                                                       |
|   | Ozone                                               | Parameterization of  |                                                       |
|   |                                                     | solar-related        |                                                       |
|   |                                                     | variations           |                                                       |
|   | Tropospheric                                        | Methodology same as  |                                                       |
|   | aerosols                                            | PiControl            |                                                       |
|   | Vegetation                                          | Methodology same as  |                                                       |
|   |                                                     | PiControl            |                                                       |
|   | Land-cover changes                                  | Same data set as     | Lawrence et al., 2016; Hurtt et al., in prep.         |
|   |                                                     | historical           | LUH2: http://luh.umd.edu/                      |
|   |                                                     |                      | https://pcmdi.llnl.gov/search/input4mips/             |
| 5 |                                                     |                      |                                                       |

**Table 2:** Summary of boundary conditions for the PMIP4/CMIP6 "tier-1" past1000 experiment.

 1For models that need aerosol optical properties as forcing.

10